# Global relevance of atmospheric and land surface drivers for hot temperature extremes

Yigit Uckan[1,2], Melissa Ruiz-Vásquez[1], Kelley De Polt[1,3], René Orth[1]

[1]Department of Biogeochemical Integration, Max Planck Institute for Biogeochemistry, 07745 Jena, Germany
[2]School of Integrated Climate and Earth System Sciences, University of Hamburg, 20144 Hamburg, Germany
[3]Institute for Environmental Studies, Vrije Universiteit Amsterdam, Amsterdam, 1081 hv, the Netherlands

*Correspondence to*: Melissa Ruiz-Vásquez (melissa.ruiz-vasquez@ecoclim.uni-freiburg.de)

**Abstract.** Hot temperature extremes have severe impacts on society and ecosystems. These extremes are driven by both atmospheric and land surface processes such as advection or reduced evaporative cooling. The contributions of the individual
drivers to the formation and evolution of hot extremes have been analyzed in case studies for major past events, but the global relevance of drivers still remains unclear. In this study, we determine the relevance of (i) atmospheric drivers such as wind, geopotential height, horizontal geopotential height differences and surface net radiation, as well as (ii) land surface drivers such as evaporative fraction and enhanced vegetation index for hot extremes across the globe using observation-based data. Hot extremes are identified at daily and weekly time scales through the highest absolute temperature, and the relevance of the
considered drivers is determined with an analogue-based approach. Thereby, temperature anomalies are analyzed from situations with driver values similar to that of the hot extreme. The results show that geopotential height at 500 hPa is overall the most relevant driver of hot extremes across the globe at both time scales. Surface net radiation and evaporative fraction are the second most relevant drivers in many regions at the daily time scale, while wind is the second most relevant at the weekly time scale. Regional variations in the relevance of individual drivers are largely explained by different climate regimes.
Revealing key regions and influential time scales of land surface drivers on hot extremes can inform more efficient prediction and management of the increasing threat these extremes pose.

## 1 Introduction

Hot extremes are severe weather events characterized by prolonged periods of excessively high temperatures. These events pose significant risks to human health, agriculture, ecosystems, and infrastructure, making the understanding of their drivers a
critical area of research (Anderegg et al., 2012, Goulart et al., 2021, Anderson and Bell, 2011, McEvoy et al., 2012). Moreover, because of the increasing trend of global temperatures, hot extremes have become longer, more frequent, and more intense in recent years (Seneviratne et al., 2023).

Hot extremes are often found to be linked to the atmospheric circulation anomalies. For example, studies by Woollings et al. (2018) and Brunner et al. (2017) highlight the importance of blocking systems and jet stream anomalies, for the onset and
development of these events especially in mid-latitudes. These quasi-stationary high-pressure systems disturb the westerly

flow for several days to weeks, leading to prolonged extreme surface temperatures (Brunner et al., 2017; Kautz et al., 2022). For instance, Wehrli et al. (2019) demonstrate the importance of atmospheric circulation as a key driver that trapped warm air masses and exacerbated surface heating during the 2010 Russian Heatwave. Similarly, persistent atmospheric pressure patterns were one of the key drivers of the 2003 European Heatwave (Miralles et al., 2014).

On the other hand, land surface feedback mechanisms, including evaporative cooling deficits and vegetation water stress due to low soil moisture can exacerbate the hot extremes and lead to multi-hazard events (Wulff and Domeisen, 2019, Teuling et al., 2010, Miralles et al., 2014, Hauser et al., 2016). Similarly, a study by Benson and Dirmeyer (2021) identified a critical "soil moisture breakpoint", below which the probability of heatwaves increases due to a shift in surface energy fluxes from latent to sensible heat. This sensitivity becomes even more pronounced as soil moisture approaches the "permanent wilting

point", where vegetation can no longer draw water from the soil, leading to a substantial increase in local surface temperatures. As a result, the sensitivity to soil moisture deficits significantly contributes to the severity of heat events (Dirmeyer et al., 2021). This effect underscores the spatial variability of soil moisture–temperature feedback mechanisms across different climatic zones. Specifically, transitional regions where latent heat flux strongly depends on soil moisture, exhibit more pronounced land-atmosphere coupling (Wehrli et al., 2019; Koster, 2004).

However, a joint and comparative assessment of these drivers is lacking such that also the relative importance of the land surface compared to that of atmospheric drivers is unclear (Perkins, 2015). Moreover, the drivers and underlying processes leading to hot extremes have mostly been studied in regional case studies to identify the drivers of specific events (Wehrli et al., 2019, Fischer et al., 2007). Hence, a global analysis to complement and reconcile the existing regional studies is missing (Sillmann et al., 2017). And there is no one commonly accepted definition of hot extremes such that previous research has

employed different temperature metrics or related indices, and at different time scales (Perkins and Alexander, 2013, Brunner et al., 2017, Raha and Ghosh, 2020). All this limits our understanding regarding the physical mechanisms leading to hot extremes across the globe and our skill to forecast these events.

We aim to address these knowledge gaps here by conducting a global analysis where we determine and compare the relevance of atmospheric and land surface drivers of hot extremes. This includes the identification of relevant spatial patterns and regions

of particular interest for each considered driver variable. In this context, we define hot extremes through the highest absolute temperatures and consider daily and weekly time scales. Focusing on different time scales allows us to reveal to which extent the drivers of hot extremes, as well as the spatial patterns of their relevance, change with different durations of the events. For this purpose, we employ an analogue approach to identify physical mechanisms leading to hot extremes. In particular, we focus on two distinct types of analogues, (i) we use flow analogues, which have been widely studied in the literature (Jézéquel

et al., 2018; Yiou et al., 2014, 2007) and are based on atmospheric circulation patterns that closely resemble the flow conditions of selected hot extremes. (ii) As a novel aspect of our work, we introduce land surface analogues, which are derived from land surface conditions such as vegetation states and energy balance at the surface which are similar to the respective conditions in a selected hot extreme. By integrating land analogues into our analysis, we aim to explore how land surface processes contribute to the development and persistence of heatwaves, offering a complementary perspective to the traditional focus on

atmospheric dynamics. Consequently, this can yield insights into potential differences of the mechanisms underlying the formation of hot extremes across time scales and regions.

## 2 Data and Methods

### 2.1 Considered drivers of hot extremes

The considered land-surface and atmospheric drivers of hot extremes in this study are selected based on the existence of
plausible physical pathways through which they can affect near-surface temperature (Table 1). Atmospheric variables include wind speed and geopotential height at three different atmospheric pressure levels which influence the distribution and movement of heat within the atmosphere (Xoplaki et al., 2003). Specifically, we analyze geopotential height and wind at three atmospheric levels: surface, 850 hPa, and 500 hPa. The selection of these three levels is based on their relevance to hot extremes formation and evolution, as well as findings from existing literature (Jézéquel et al., 2018). The surface level can provide
information about the advection of warm air (Jiménez-Esteve and Domeisen, 2022). The 850 hPa level, situated approximately 1.5 km above sea level, is used to assess lower-tropospheric processes. The 500 hPa level, roughly 5.5 km above sea level, is often related to hot extremes formation due to blocking mechanisms at this level (Zschenderlein et al., 2019). This level is important for capturing mid-tropospheric patterns and the influence of large-scale atmospheric circulation on weather systems (Ventura et al., 2023). In addition, we use horizontal geopotential height differences at 500 hPa as a proxy for the geostrophic
wind.

Land surface variables, such as evaporative fraction (EF), enhanced vegetation index (EVI), and surface net radiation, impact near-surface temperatures through the provision of energy, evaporative cooling as well as albedo which determines the amount of reflected solar energy (Seneviratne et al. 2010). We analyze the land surface variables across the same time scales considered for the hot extremes, but also at longer time scales in order to capture potential lagged effects arising from an accumulation of
the influence of land surface variables over time, given their relatively slower variability compared to the atmospheric drivers. These time scales are important for understanding plant responses to hot extremes: water loss and stomatal closure are more pronounced on the daily time scale, while on the daily to weekly time scale vegetation is affected by leaf wilting and senescence (Zhang et al., 2016).

We consider two different time scales to define hot extremes, 1 day and 7 days. This allows to test to which extent the
underlying drivers depend on the considered time scale. The spatial and temporal resolutions considered are 0.25 degrees and daily intervals, respectively, for the study period from 2001 to 2020. This period was selected because the evapotranspiration data from the X-BASE dataset used to calculate EF is only available during these years. EF is computed by normalizing evapotranspiration by surface net radiation. Notably, this calculation incorporates variables from two different datasets: X-BASE and ERA5. This approach is justified, as X-BASE is formulated using ERA5 data. In cases where the native resolutions
of the datasets differed from the ones mentioned, the datasets were aggregated to a spatial resolution of 0.25 degrees and a daily temporal resolution using linear interpolation. Regarding the limited global coverage of the EF dataset, we mask grid

cells without data such as Greenland in all considered datasets to ensure consistency in spatial coverage. In addition, we compute the horizontal geopotential height differences at 500 hPa pressure level for each grid cell with respect to the values in adjacent grid cells in the northern, eastern, southern and western directions.

To capture potential lagged effects arising from an accumulation of the influence of land surface variables over time, we average land-surface variables at different time windows depending on the considered hot extreme time scales. For example, for 1-day hot extreme events (Table 1; third column) we use values on the day of the event, an average of the variables on the event day and the 2 preceding days, as well as an average of the variables on the event day and the 14 preceding days. Similarly for 7-day hot extremes we consider time windows of 7, 14 and 28 days to compute the average of the land surface variables.

**Table 1** Summary of considered driver variables

| Variables | Source | 1-day hot extremes | 7-day hot extremes |
|---|---|---|---|
| **Geopotential Height** | ERA5 (Hersbach et al., 2020) | Pressure at the Surface | Pressure at the Surface |
| | | Geopotential height at 850 hPa | Geopotential height at 850 hPa |
| | | Geopotential height at 500 hPa | Geopotential height at 500 hPa |
| **Wind Speed** | ERA5 (Hersbach et al., 2020) | Wind at the Surface | Wind at the Surface |
| | | Wind at 850 hPa | Wind at 850 hPa |
| | | Wind at 500 hPa | Wind at 500 hPa |
| **Geopotential Height Difference** | ERA5 (Hersbach et al., 2020) | Geopotential height difference at 500 hPa | Geopotential height difference at 500 hPa |
| **Enhanced Vegetation Index (EVI)** | MODIS (Didan, 2015) | EVI 1-day | EVI 7-day |
| | | EVI 3-day | EVI 14-day |
| | | EVI 15-day | EVI 28-day |
| **Evaporative Fraction (EF)** | X-Base (Nelson et al., 2024) ERA5 (Hersbach et al., 2020) | EF 1-day | EF 7-day |
| | | EF 3-day | EF 14-day |
| | | EF 15-day | EF 28-day |
| **Surface Net Radiation** | ERA5 (Hersbach et al., 2020) | Radiation 1-day | Radiation 7-day |
| | | Radiation 3-day | Radiation 14-day |

Radiation 15-day                    Radiation 28-day

## 2.2 Definition of hot extreme events

For our definition of hot extremes, we use the 2 m daily mean temperature as this accounts for both day-time and night-time
conditions. We make this choice because night-time temperatures are relevant for impacts of hot extremes on human health, they play a role in the physiological response of plants to hot extremes (Wahid et al., 2007), and because this way we can also capture trends in night-time extreme temperatures (Wu et al., 2023). These daily means used in our analysis are computed based on UTC instead of local time. While this choice may lead to phase mismatches in diurnal cycles for some variables, particularly in regions where local time differs significantly from UTC, it provides consistency across datasets, which is
essential for our analysis.

We identify the hot extreme events in each grid cell based on the highest absolute temperature values within our study period from 2001 to 2020. The selected events occur during the warm seasons, as we pick the events with the highest temperatures. For the 1-day hot extreme events, we select three individual days with the next highest temperatures, ensuring that each selected day is at least 15 days apart from the others to maintain independence, as shown in Fig. 1(a). Likewise, for the 7-day time scale
we select the three 7-day periods with the next highest average temperatures, also ensuring that they are at least 15 days apart from each other for independence. In order to further illustrate the event selection of 1-day hot extremes within the study period (2001- 2020), let's assume in a specific grid cell, the hottest day recorded during this period is July 15, 2012. After selecting this day, we mask out this July 15 and the 30 days surrounding it (July 1 to July 30) to prevent selecting any overlapping or consecutive days in the further selection process. We then identify the second hottest day from the remaining days of the time
series after the masking, which could be August 5, 2010, and apply the same 15-day masking around this date (July 21 to August 20). This process is repeated to find the third hottest day, ensuring that all three selected days are at least 15 days apart, maintaining their independence.

For 7-day hot extreme events, the procedure is similar. Suppose the highest 7-day average temperature in the grid cell occurs from July 10 to July 16, 2015. We mask out this period and the surrounding 30 days (June 26 to August 1) to select the next
highest 7-day period, such as August 20 to August 26, 2013. This ensures that each selected 7-day event is independent.

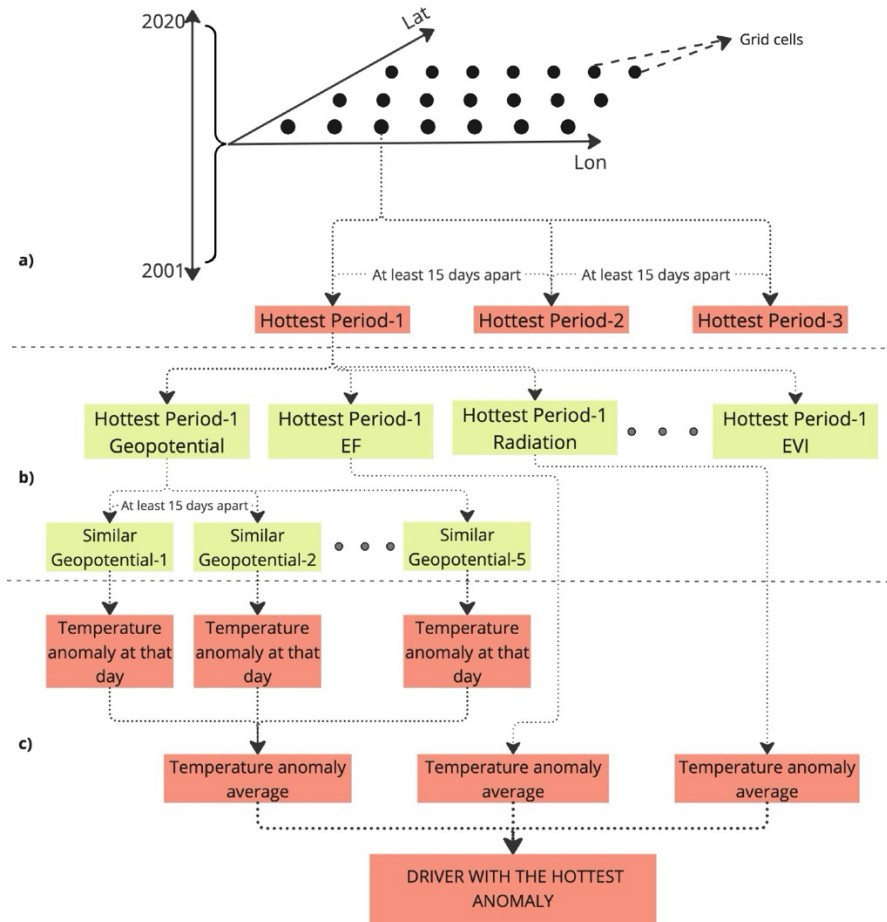

**Figure 1** Workflow for determining main drivers of hot temperature extremes. Hottest periods refer to 1-day or 7-day hot extreme events. See text for details. (This figure is created by using miro.com)

## 2.3 Description of the analogue approach

After identifying hot extremes, we determine the values of the considered driver variables for each event and based on them we identify analogues as illustrated in Fig. 1(b). For each driver and at each considered atmospheric level (in the case of geopotential height and wind) and temporal scale (in the case of EVI, EF, and surface net radiation) we select the five periods in which the driver variable values are most similar to those observed during the identified hot extremes. This approach shares a conceptual basis with the analogue methods in the literature, such as those used by Jézéquel et al. (2018) and Yiou et al.

(2007). These studies show that selecting more than five geopotential height analogues has little effect on the results. To maintain consistency, we use the same number of five analogues for each driver variable. The analogue analysis is done separately for each grid cell, i.e. analogues always come from the same grid cell where the hot extreme was detected. These analogues are only selected within a similar time of the year as the original hot extremes event to ensure comparable conditions;

this is needed as e.g. circulation patterns with westerly winds from the North Atlantic tend to cool Europe in summer while
warming it in winter, or because EVI or EF anomalies are only comparable across a similar phase of the growing season. For
this purpose, a 120-day window surrounding the specific calendar date (i.e., month and day) of the relevant hot extreme event
is considered across all years to select the analogue periods. These selected periods are also at least 15 days apart from each
other to ensure independence.

We note that the "closeness" of the selected analogues to the observed values on the day/week of the event can influence our
results. In general, the analogues resemble the extremes well (Fig. A1). They tend to be closer to the observed values for the
1-day extremes than the analogues for the 7-day extremes. Analogues for radiation in the high northern latitudes are most
different from the extremes with an underestimation of the observed radiation of 20% or more locally. This makes it less likely
to detect radiation as a main driver in these regions.

**2.4 Degree of relevance index**

For each driver, we compute a degree of relevance which describes the fraction of the observed temperature anomaly of the
selected hot extremes which is detected in the related analogues for each driver variable, as shown in Fig. 1(c).

For each variable $D$ (e.g., geopotential height, EVI, EF), we identify 15 analogue periods based on similarity to the three hottest
observed extremes (5 analogues each) in each grid cell. The degree of relevance for each variable "D" in each grid cell "g" is
computed as Eq. (1):

$$Degree\ of\ relevance(g, D) = \frac{1}{15}\sum_{i=1}^{15}\left(\frac{T'_{analog}(g,i)}{\bar{T}'_{event}(g)}\right) \tag{1}$$

$T'_{analog}(g, i)$ denotes the temperature anomaly (climatological mean temperature subtracted from the temperature values
recorded) during the i-th analogue period in grid cell g, based on the conditions of driver D. $\bar{T}'_{event}(g)$ is the mean temperature
anomaly calculated from the three observed hottest extreme events in the grid cell g. It serves as a basis of comparison to
determine how much of the observed extreme temperature anomaly can be explained by the analogue temperature conditions
of variable D. This indicates the fraction of the actual temperature anomaly that can be explained with one driver, i.e. the
degree of relevance of the dominant driver in the considered hot extreme events. The expected degree of relevance ranges
between 0 and 1. Values close to 1 indicate that hot extremes are explained to a great extent by the considered variable. The
dominant driver in a grid cell is then selected as the variable with the highest degree of relevance. It is important to note that
this approach assumes a linear and separate contribution of each driver, which is a limitation when interactions between drivers
are relevant. We note that with our approach the cumulative degree of relevance from different drivers can exceed 1 due to
collinearities among the Earth system variables. However, as we explain in more detail in Section 3.2, this is typically not the
case

## 2.5 Analysis of driver independence

In our methodology (sections 2.3 and 2.4), we assume that atmospheric and land-surface variables are independent as drivers of hot extremes. In order to assess the extent of their independence, we compute cross-correlations among the driver variables across several regions worldwide. This ensures to avoid that relationships cancel out between different regions. We focus on 5×5 grid cell (1.25°×1.25°) clusters among regions with different climatic and land-surface characteristics (i.e., Central Europe, the Amazon rainforest, Australia, Central Asia, South Africa and North America). To avoid potential biases from a single cluster per region, we select three separate 5×5 subregions, thus capturing a broader spectrum of climatic and topographic conditions (Fig. A2). Within each subregion's 25 grid cells, we have three hot extreme events per grid cell (as mentioned in Section 2.2). This results with 5×5×3 = 75 observations per subregion for calculating the correlations between variables. By averaging the correlation results across the three subregions for each region (75×3 = 225 observations), we obtain an aggregated indicator of the correlation of variables).

## 2.6 Attribution Analysis

In order to analyze the spatial distribution of the dominant driver variables identified for 1-day and 7-day hot extremes with respect to different land surface characteristics and climatic regimes, we employ a random forest approach (Breiman, 2001; Molnar, 2020). Therein, the spatial patterns of the degree of relevance of geopotential height and EF serve as target variables while aridity, tree cover, irrigation, silt soil fraction, 2 m temperature, total precipitation, surface net radiation, soil moisture and topography are used as predictors (Fig. A3). We split the data into training and testing sets, with 25% of the data reserved for testing. We used 100 trees, and a maximum depth of 10 to configure the RandomForestRegressor, as these hyperparameters have proved to work well in other studies (Oshiro et al.; 2012; Probst and Boulesteix 2017). Bootstrapping was enabled, and the feature importance was evaluated using mean absolute SHAP (Shapley Additive Explanations) values (Lundberg and Lee, 2017; Sundararajan and Najmi, 2020) to provide insight into the contribution of each predictor.

## 2.7 Effect of the positive trend in hot temperature extremes on the relevance of driver variables

In the light of the increasing trends in global temperature extremes (Seneviratne et al., 2023), we analyze potential changes in the relevance of the considered drivers of hot extremes over time. For this purpose, we divide the study period into two periods, 2001-2010 and 2011-2020, and employ the same methodology as described in Sections 2.1 to 2.4 to calculate the relevance of all driver variables for each of the time periods. We also determine the significance of the changes in relevance through bootstrapping. For this purpose, we use the original 15 analogues (3 hot extremes x 5 analogues per hot extremes = 15 analogues) per grid cell and per time period (2001-2010 and 2011-2020), and draw a random sample of 15 from them (with replacement). Resampling is done 1000 times. Then, we compare the mean degrees of relevance of the resampled 15 analogues from both periods per grid cell such that we can infer significance from analyzing whether a substantial fraction (e.g. 950 out of the 1000) of the mean degrees of relevance between the two time periods is consistently above or below zero.

# 3 Results & Discussion

## 3.1 Global distribution of most relevant driver variables

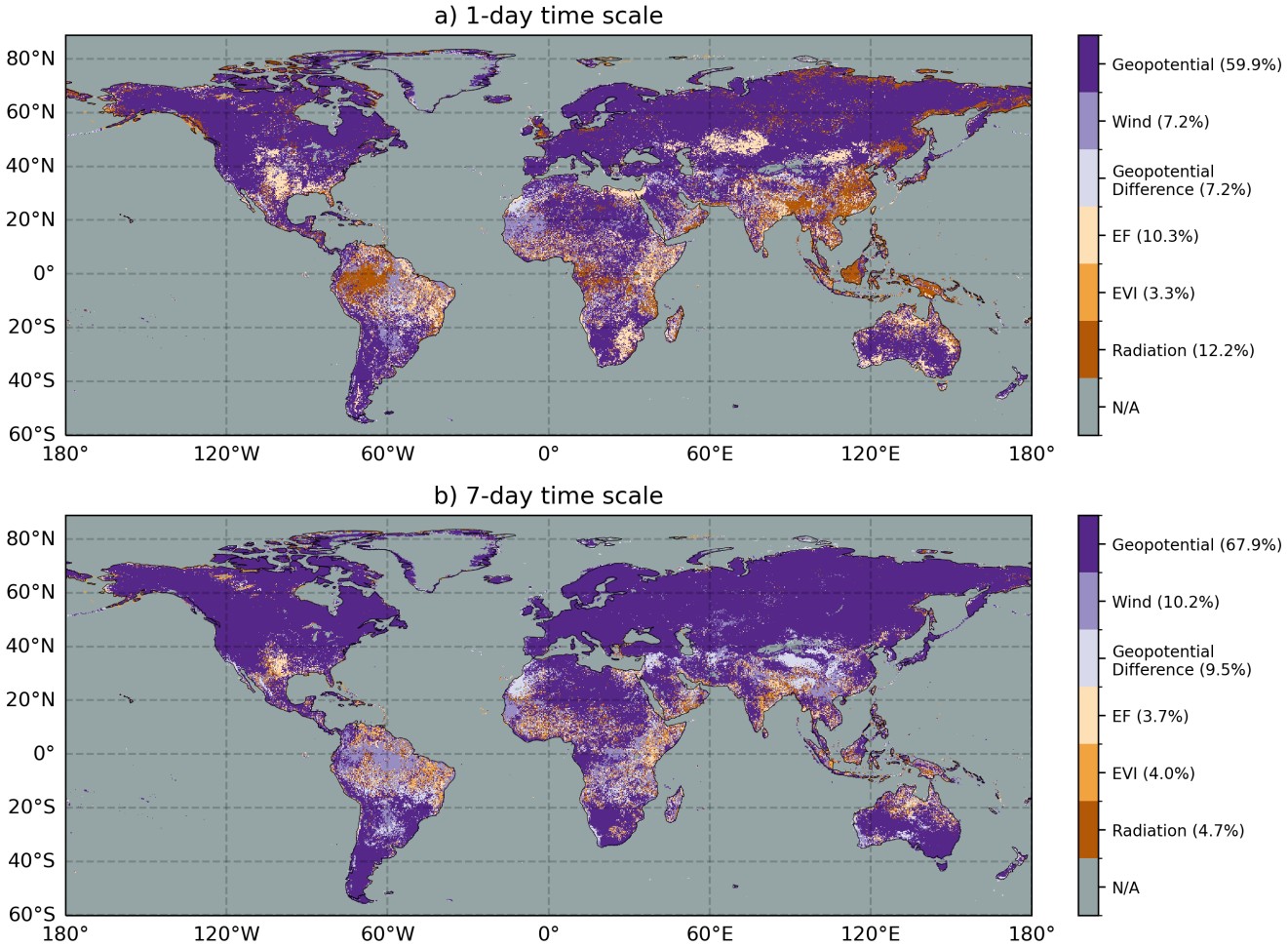

**Figure 2** Dominant driver variables identified for 1-day and 7-day hot extremes. The percentages indicated on the color bar reflect the proportion of the study area where hot extremes are most influenced by each variable

The global distribution of the dominant driver variables for both 1-day and 7-day time scale hot extremes are illustrated in Fig. 2. A more detailed depiction of drivers' relevances across pressure levels and time scales is presented in Fig. A4. The results are aggregated across geopotential height levels in the case of the atmospheric variables, and across time scales in the case of the land surface variables. The map corresponding to the 1-day time scale reveals that geopotential height is the predominant driver, accounting for approximately 60% of the analyzed area. More specifically, the 500 hPa level geopotential height is most influential in mid-latitude regions. This finding supports existing literature that highlights the significant role of atmospheric blocking mechanisms in the formation of hot extremes in these latitudes and at 500 hPa pressure level (Pfahl and

Wernli, 2012; Brunner et al., 2017; Jiménez-Esteve and Domeisen, 2022). Conversely, in primarily tropical regions, surface net radiation substantially influences the occurrence of 1-day extreme temperature events. This can be understood as lateral temperature and pressure gradients are weaker in the tropics such that atmospheric circulation is less relevant while instead solar radiation is intense because of a larger and near-direct solar incident angle. Next to this, EF is found to be the most relevant hot extremes driver across 10 % of the study area. This is particularly the case in Central North America and Central Asia which are known as transition regions between wet and dry climate. In these regions evapotranspiration and consequently evaporative cooling is relatively high but also limited by soil water availability which is typically low during hot extremes (Koster, 2004; Wang et al., 2007). This way, reduced soil moisture and, as a result, lower than usual EF leads to higher sensible heat flux, resulting in increased surface temperatures during hot extremes (Schwingshackl et al., 2017; Teng et al., 2016). Similarly, the findings of Röthlisberger and Papritz (2023) show that diabatic heating is a factor affecting the temperature anomalies especially in regions where the soil moisture is limited.

For drivers of hot extremes at the 7-day time scale we find that geopotential height remains the most dominant variable globally for 7-day hot extremes, while its relative influence increases with respect to the 1-day hot extremes in mid-latitudes. This can be related to the formation of blocking systems during which a stationary high-pressure system blocks the westerly flow at midlatitudes (Brunner et al., 2017), causing clear-sky radiative forcing, subsidence of air masses and air warming (McGregor, 2024). In the case of the land surface-related variables, particularly EVI affects hot temperatures through shading in tropical or semi-arid warm regions. In the identified regions where EVI is most relevant through the vegetation response to incoming solar radiation and emission of longwave radiation (McPherson, 2007), temperatures can reach levels at which leaves start to wilt which then reduce shading-related cooling and further amplifies the temperatures (Brun et al., 2020). This indicates that heat can accumulate through higher than usual radiation persisting over some time (Miralles et al. 2014). Furthermore, land surface variables affect hot extremes mostly at the same time scale of the hot extremes while lagged effects are of minor relevance (Fig. A4, Table A1).

The main driving variables of hot extremes are summarized in Fig. 3 with respect to climate regime as classified through long-term means of temperature and aridity (calculated as the ratio of long-term mean surface net radiation and long-term mean daily accumulated precipitation). The choice of temperature and aridity in this context is justified with an additional analysis in section 3.2 and Figure A3. Geopotential height is the most relevant driver across both cold and warm regions as well as dry and wet climate regimes for both 1-day and 7-day time scales. At the 1-day time scale, in warm regions, radiation is the second most relevant driver which is related to more intense solar radiation in low-to-medium latitudes. The land surface variable EF is the second most important driver in dry climate regimes. This is related to the fact that water availability is just about sufficient for vegetation in these regions which means that (i) it can supply significant evaporative cooling while (ii) during warm and dry conditions water availability will not be sufficient such that evaporative cooling decreases which in turn contributes to enhanced temperatures. For the 7-day time scale, wind becomes relevant for warm regions next to geopotential, which probably relates to the advection of warm air from remote regions (Domeisen et al., 2022; Zschenderlein et al., 2019), while radiation and EF are less relevant.

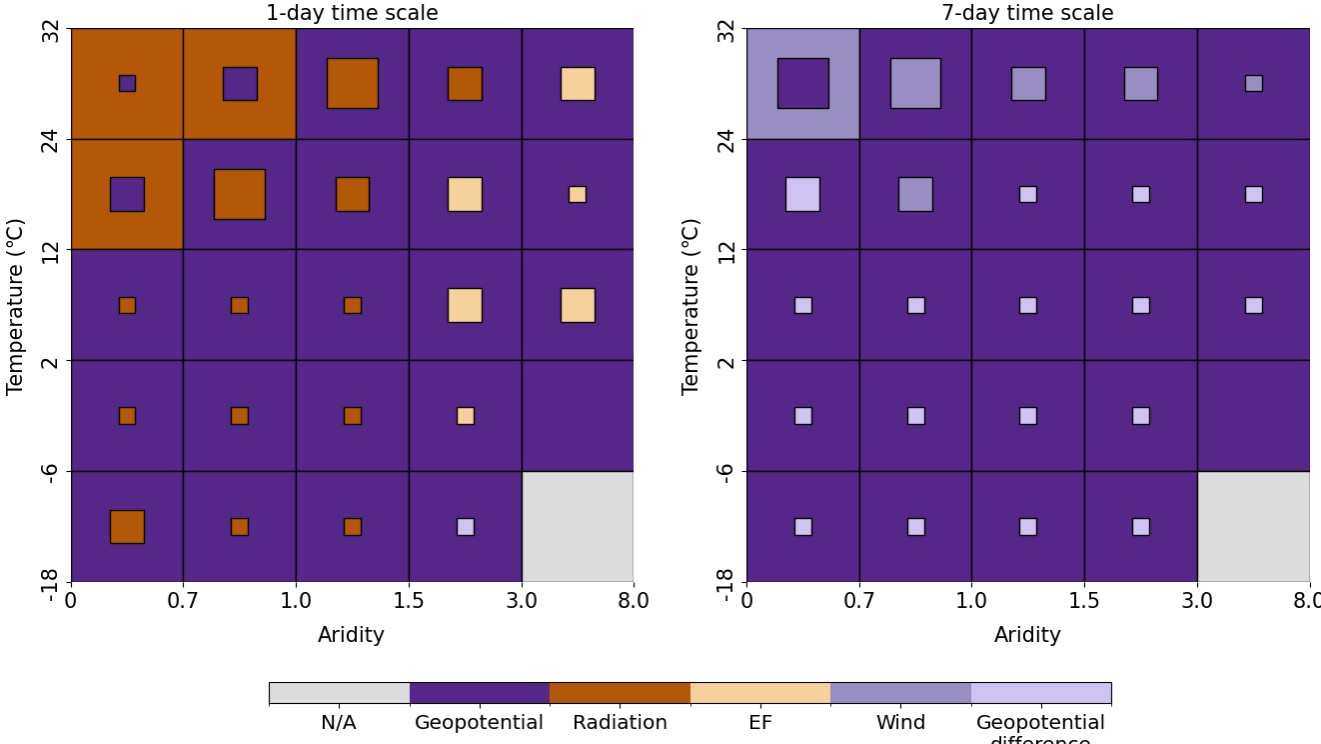

**Figure 3** Main driving variables of hot extremes summarized across climate classes. Color of the boxes indicates which driver is most influential in the largest number of grid cells within the climate class. Color of the inner square indicates the second most relevant driver. The size of the square denotes the relative relevance of the second most important driver where large squares are used if the number of grid cells where the second most relevant driver is most influential exceeds 65 % of the number of grid cells where the most relevant driver is most influential. Likewise, medium-sized squares are used for fractions of 25 % - 65 % and small squares in the case of <25 %. Note that if there are fewer than 20 grid cells to represent the corresponding variables, the boxes will appear in gray or any other single color. The latter is the case when there are sufficient grid cells of the most dominant variable but an insufficient number of grid cells for the second-most dominant variable. The number of grid cells in each category is shown in Fig. A5.

These results have to be seen in the light of some limitations. The analogue methodology employed here does not account for interactions between the considered driver variables. This means that temperature anomalies associated with the analogues of a given variable could also partly result from anomalies in another variable that is closely connected to the first one and will hence add to its effect. We expect that the consideration of several hot extremes and of multiple analogues for each extreme can mitigate this problem as different weather and vegetation conditions characterize each of them. Furthermore, our main goal is to disentangle land surface and atmospheric drivers of hot extremes which are not expected to be strongly related to each other. This assumption is tested by performing a cross-correlation analysis of the variables for each selected region (see Section 2.5). Overall, we find that most correlations are below 0.5 (=25% of shared variance) with some exceptions. While some moderate correlations exist within atmospheric variables or land surface variables, correlation of land and atmospheric variables is low for most considered regions (Fig. A6).

Another limitation is the data quality of each driver variable. A lower signal-to-noise ratio for certain variables compared to others may affect the identification of analogues and related temperature anomalies, and consequently the estimated relevance of the variable. However, we observe consistent spatial patterns across different datasets which enhance credibility to our results and align with the existing literature on land surface and atmospheric drivers for hot extremes (Jimenez-Esteve and Domeisen, 2022). At the same time, we use established products which are all comprehensively validated and hence we expect

that differences in data quality between individual data streams do not have an effect on our results.

## 3.2 Relative roles of the most important atmospheric and land surface drivers

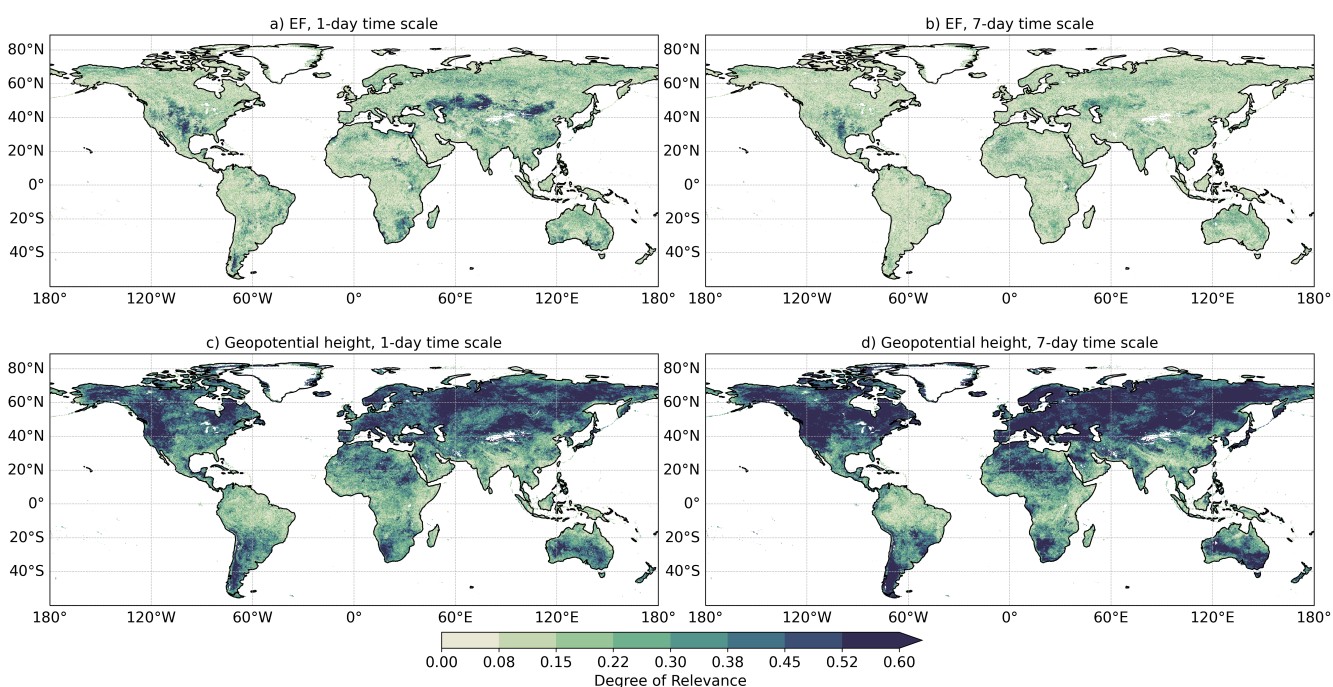

**Figure 4** Degree of relevance of EF (a, b) and geopotential height (c, d) at daily and weekly time scales. The degree of relevance is computed as the ratio between the respective analogue temperature anomalies and the observed temperature anomalies during hot extremes.

In this section we analyze the main land surface and atmospheric drivers in more detail in terms of their degree of relevance. The latter is calculated as the fraction of temperature anomalies of hot extremes explained by each driver according to the temperature anomalies of their analogues. Fig. 4 presents the results for EF and geopotential height for 1-day and 7-day hot extremes. The relevance of EF diminishes from daily to weekly timescales, suggesting that vegetation influences on extreme heat primarily operate over short durations with immediate effects. This is likely because, at longer timescales, the evaporative

cooling effect becomes less dominant as atmospheric advection transports air masses from other regions, overshadowing local land-atmosphere interactions. Vice versa, the relevance of geopotential height also increases from daily to weekly time scales, especially in the mid-latitudes, coinciding with the regions where Pfahl and Wernli (2012) found that atmospheric circulation was highly related to extreme summer temperatures. This can be related to the persistence of the blocking systems for several

days to weeks (Brunner et al., 2017), exacerbating the subsidence of air masses and air warming for longer time periods (McGregor, 2024).

In the next step, we are determining the drivers of the spatial patterns of the relevance of geopotential height and of EF in supporting hot extremes. This is done with a random forest analysis (section 2.6). We find that long-term mean temperature and radiation are the most relevant predictor variables for both 1-day and 7-day hot extremes (Fig. A3). Additionally, aridity and topography (based on the standard deviation of subgrid altitude values) play an important role for both time scales of hot extreme events while the other considered variables are less important.

In the following we group the global results with respect to long-term averages of temperature and aridity; temperature is chosen as it is found to be the most relevant variable, and aridity is chosen instead of radiation because it covers another aspect of climate (e.g. including water input to the land surface) while radiation is more linked with temperature. Accordingly, we summarize the results from the global degree of relevance of EF and geopotential height in different climatic regions (Fig. A7). Although, EF is the second most dominant variable (after geopotential height) in semi-arid and arid regions as shown in Fig. 3, EF has a relatively higher degree of relevance in those regions compared to wet regions. On the other hand, geopotential height is more relevant in colder climates. These findings highlight that climate is the main modulator of the relevance of drivers of hot extremes across regions.

Furthermore, we study potential changes in the relevance of the considered drivers of hot extremes between the periods 2001-2010 and 2011-2020 (Fig. A8). We find only small changes. While these could be related to natural decadal variability, we find that radiation and EVI are becoming slightly more relevant at both considered time scales, which may be also related to global greening (Zhu et al., 2016, Chen et al., 2019) and global brightening (Wild, 2009). At the same time, the changes are significant only in relatively small fractions (<10% for most drivers) of the study area.

Moreover, we calculate the sum of the degree of relevance of the three most influential variables at each grid cell (Fig. A9). This indicates which fraction of the observed hot temperature anomalies can be explained with our approach. The maps show that a large part of the observed hot temperature anomalies can be explained within the analogue approach. This suggests that we have considered relevant and meaningful driver variables. The explained fractions of hot extreme temperature anomalies are slightly lower in some tropical and subtropical regions. This can be related to lower quality of land surface datasets due to sparse observations and partly dense vegetation cover, as well as to the influence of sea surface temperatures on hot extremes in these regions (Orth and Seneviratne, 2017). Note that the consideration of sea surface temperatures or other remote forcings on hot extremes is outside the scope of this study. Correspondingly, we also do not focus on time scales longer than 7 days. Note further that in this calculation we do not jointly consider variables which are related to each other, i.e. geopotential height and geopotential height difference, or EVI and EF, such that we mitigate the effect of collinearities among the considered drivers. If such related variables are among the three most influential variables, we consider instead the next most relevant variable which is less related to the already considered variables. In some cases, the cumulative degree of relevance of the variables could exceed a value of 1 due to collinearities among driver variables. Typically, however, this is not the case as shown in Figures 4 and A9. This indicates that dependencies between driver variables are not critically affecting our analysis.

## 4 Conclusions

This study provides a global analysis of the potential drivers of hot extremes, considering a selection of atmospheric and land surface variables. The results highlight that geopotential height, particularly at the 500 hPa level, is globally the dominant driver for hot extremes, and especially in mid-latitude regions. This finding underscores the important role of atmospheric circulation anomalies, such as atmospheric blocking, in the formation of hot extremes (Pfahl and Wernli, 2012). In contrast, surface net radiation is more influential in tropical regions, where it is more intense and constant throughout the year and can therefore exacerbate hot conditions at any time. Land surface variables, like evaporative fraction and enhanced vegetation index, influence hot extremes in transitional regions (semi-arid to arid) which are neither wet nor dry such that they can sustain significant evapotranspiration. Moreover, evapotranspiration depends on water availability such that soil moisture (i.e. a land surface variable) variability influences evapotranspiration and consequently the surface energy balance and temperature (Denissen et al., 2024; Seneviratne et al., 2010).

These results complement the existing literature on drivers of hot extremes by jointly considering and comparing the relevance of atmospheric versus land surface drivers which were so far largely studied in isolation. Another novel aspect in our study is the consideration of hot extremes at different time scales. As the time scale of hot extremes is not clearly defined in the literature, we focus on daily and weekly time scales in order to analyze potential differences of drivers' relevances across time scales. By examining both 1-day and 7-day time scales, we capture different phases of heatwaves; 1-day events reflect the peak of extreme heat, while 7-day events represent persisting hot conditions. This approach allows us to infer that atmospheric drivers are slightly more relevant at longer time scales, whereas land surface drivers, such as surface drying and reduced evaporative cooling, are slightly more relevant at shorter time scales. This may be related to the fact that local evaporative cooling is overshadowed by the advection of air from other regions via the large-scale circulation at longer time scales. Another interesting result of our study is that despite the increase of hot extremes in response to global warming, there is no clear global shift in the relevance of the drivers. We find only regionally significant changes in the relevance of geopotential height, EVI and radiation. It remains unclear, however, if the small changes in driver's relevances during a relatively short 20-year study period can be interpreted as an indication of negligible changes in the mechanisms underlying hot extremes in future decades up until the end of the century.

Our findings imply that weather forecasting models and Earth System models need to be able account for various mechanisms leading to hot extremes in order to yield accurate forecasts of unfolding hot extremes as well as future projections of their occurrence and intensity. In particular, inclusion of vegetation phenology could be crucial, as variables like EVI and EF are linked to vegetation processes, such as canopy conductance and stomatal resistance, which play a significant role in driving hot extremes. However, many current forecasting models do not sufficiently exploit the available vegetation data such that they e.g. use only mean seasonal cycles instead of near-real time dynamics (Ruiz-Vásquez et al., 2023; Duveiller et al., 2022). Including these processes would improve the representation of land-atmosphere interactions, which is vital for enhancing the accuracy of hot extreme predictions. This way, our study motivates (i) further efforts to model the vegetation response to

hydro-meteorological conditions at high spatial resolution where the coupling between vegetation and weather can be most accurately represented, as well as (ii) interest to monitor the root-zone soil moisture dynamics to better constrain vertical and lateral soil water movement in land surface models such that they can yield more accurate estimates of plant-available water. Previous research has shown in a case study that hot extremes cause most impacts in terms of societal attention and public health at time scales between 2 weeks and 2 months (De Polt et al., 2023). This highlights the need for more comprehensive and multidisciplinary studies that build on our findings to investigate and compare the relevance of different drivers of hot extremes at weekly to monthly timescales. Such studies could integrate atmospheric, land surface, and oceanic influences to provide a holistic understanding of the mechanisms behind prolonged hot extremes. Considering the role of ocean variability and broader spatial patterns, including teleconnections could help reveal how large-scale climatic factors interact with local conditions to intensify or sustain extreme heat events. This broader perspective could be useful for enhancing predictive models and informing adaptive strategies in the face of increasing climate variability.

# Appendix

Table A1 A summary of a combined percentages of all variables from Fig. A4 in a tabular format. Detailed dominant driver variables identified for 1-day and 7-day hot extremes. Percentages represent the area where each driver is most influential.

| Variable (1-day hot extremes) | Percentage (%) | Variable (7-day hot extremes) | Percentage (%) |
|---|---|---|---|
| Radiation 1-day | 6.8 | Radiation 7-day | 2.0 |
| Radiation 3-day | 4.0 | Radiation 14-day | 1.4 |
| Radiation 15-day | 1.3 | Radiation 28-day | 1.3 |
| EVI 1-day | 1.3 | EVI 7-day | 1.7 |
| EVI 3-day | 0.9 | EVI 14-day | 1.2 |
| EVI 15-day | 1.0 | EVI 28-day | 1.0 |
| EF 1-day | 5.7 | EF 7-day | 1.6 |
| EF 3-day | 3.3 | EF 14-day | 1.1 |
| EF 15-day | 1.4 | EF 28-day | 1.0 |
| Geopotential diff north | 1.9 | Geopotential diff north | 3.0 |
| Geopotential diff east | 1.5 | Geopotential diff east | 1.5 |
| Geopotential diff west | 1.6 | Geopotential diff west | 1.7 |
| Geopotential diff south | 2.2 | Geopotential diff south | 3.3 |
| Wind Surface | 3.2 | Wind Surface | 5.7 |
| Wind 850 hPa | 2.9 | Wind 850 hPa | 2.8 |
| Wind 500 hPa | 1.1 | Wind 500 hPa | 1.7 |
| Pressure surface | 8.4 | Pressure surface | 5.0 |
| Geopotential 850 hPa | 2.7 | Geopotential 850 hPa | 2.4 |
| Geopotential 500 hPa | 48.8 | Geopotential 500 hPa | 60.5 |

Normalized Differences for Radiation, Geopotential, and EVI

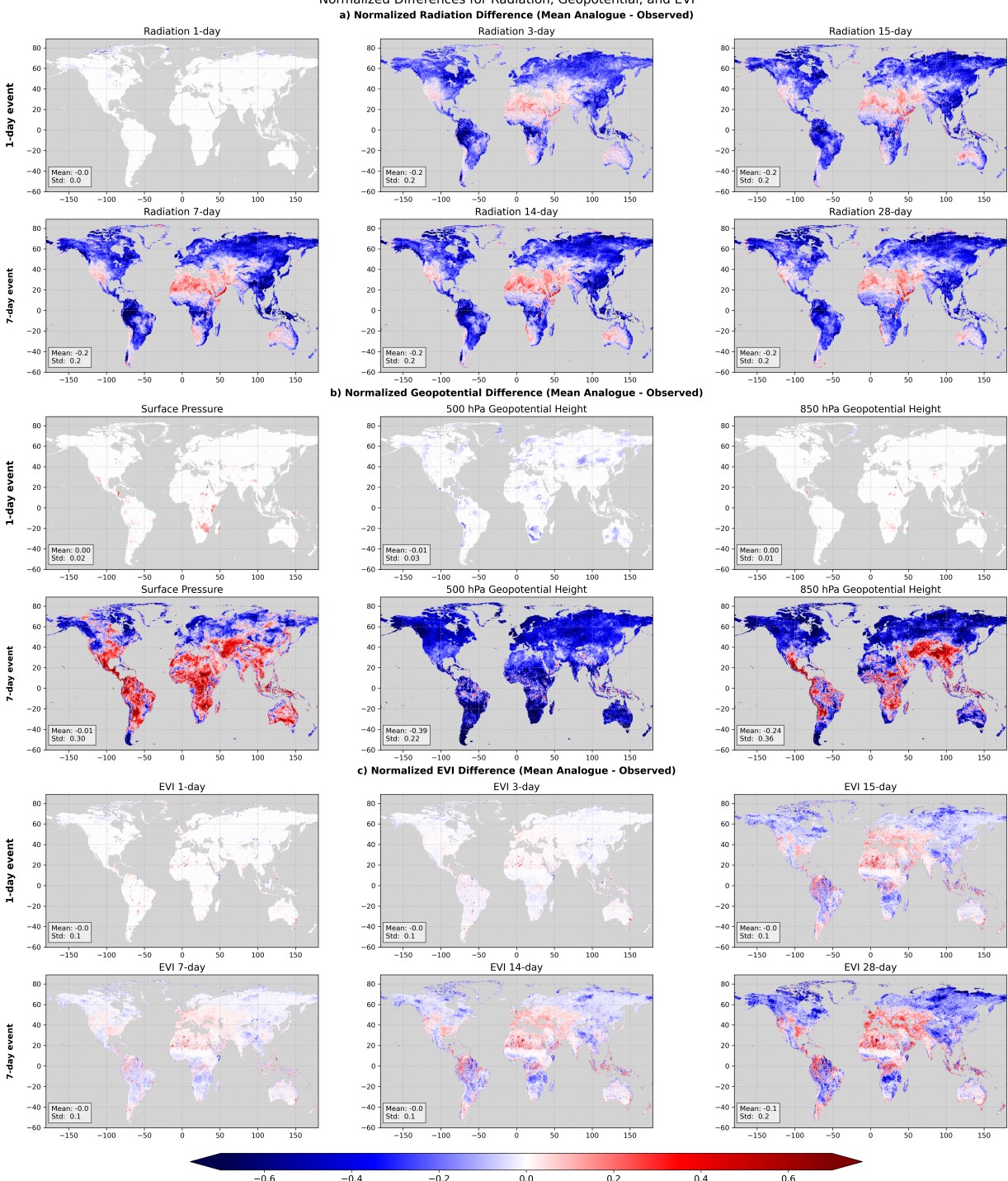

**Figure A1** Spatial patterns of normalized differences of geopotential height, radiation and EVI between the mean values of the five analogue periods and the values of the observed hot event divided by the local 20-year standard deviation. Note that this figure shows the average differences across the three hottest periods. Mean and standard deviation values are denoted in the bottom left corner of each map

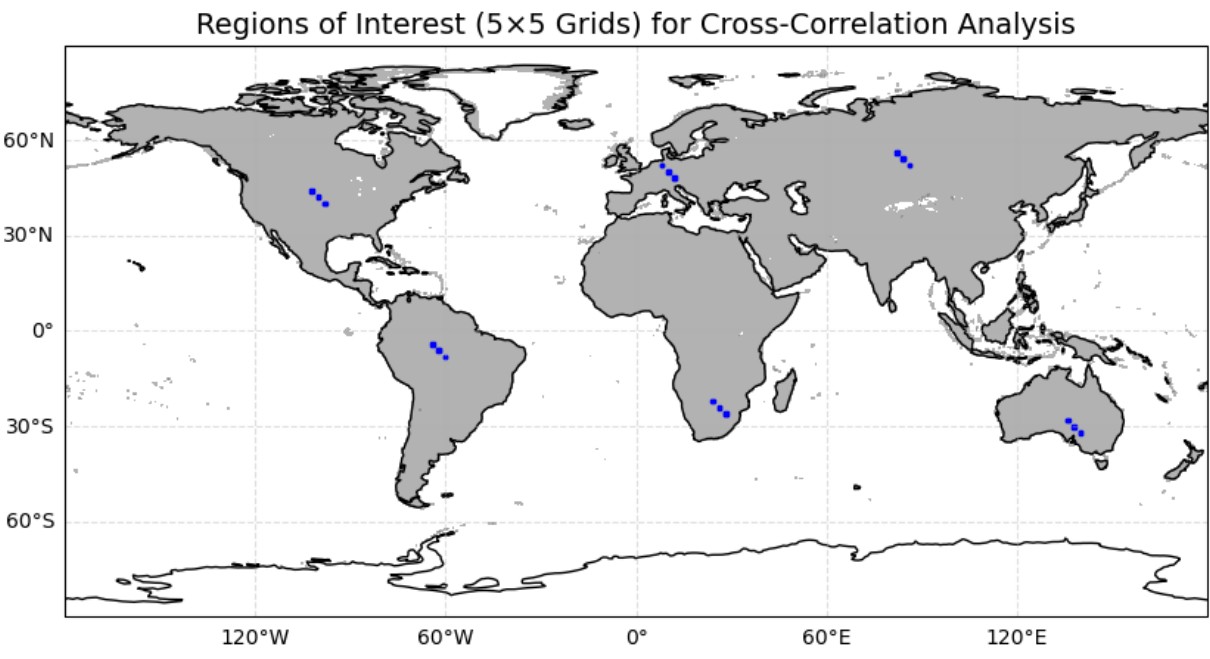

**Figure A2** Selected regions from Central Europe, the Amazon rainforest, Australia, Central Asia, South Africa and North America. Each region is divided into three subregions and within each subregion, a 5x5 grid cell cluster is used to assess the cross-correlation of driver variables during hot extreme events.

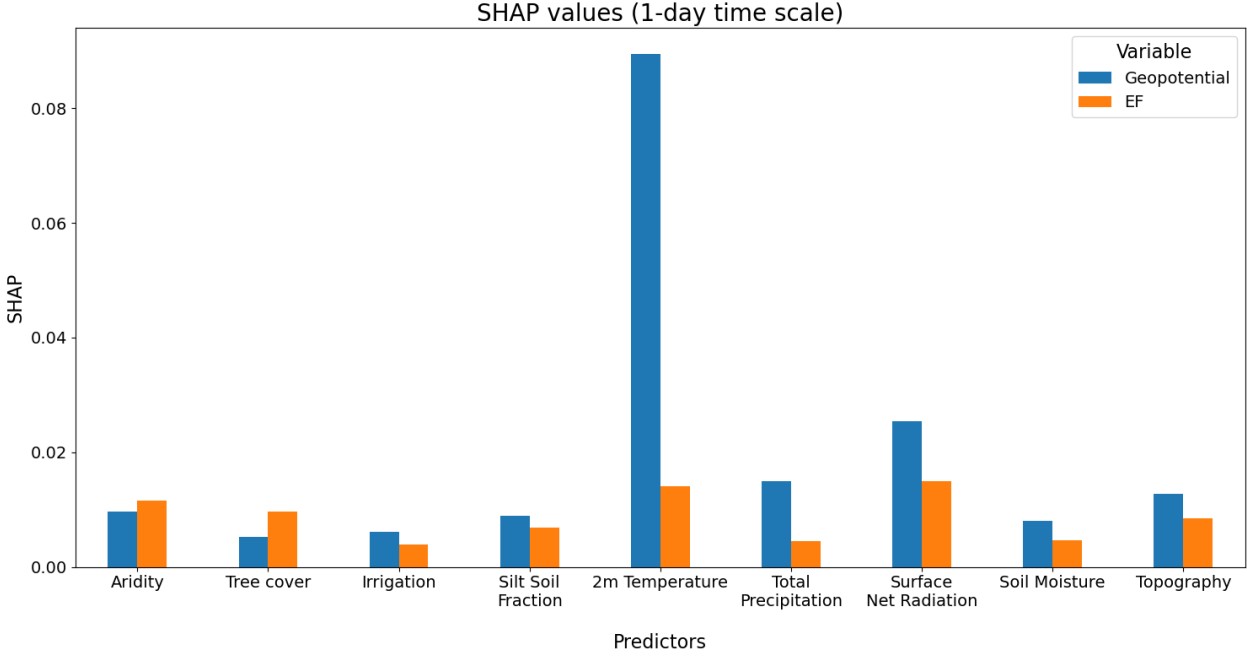

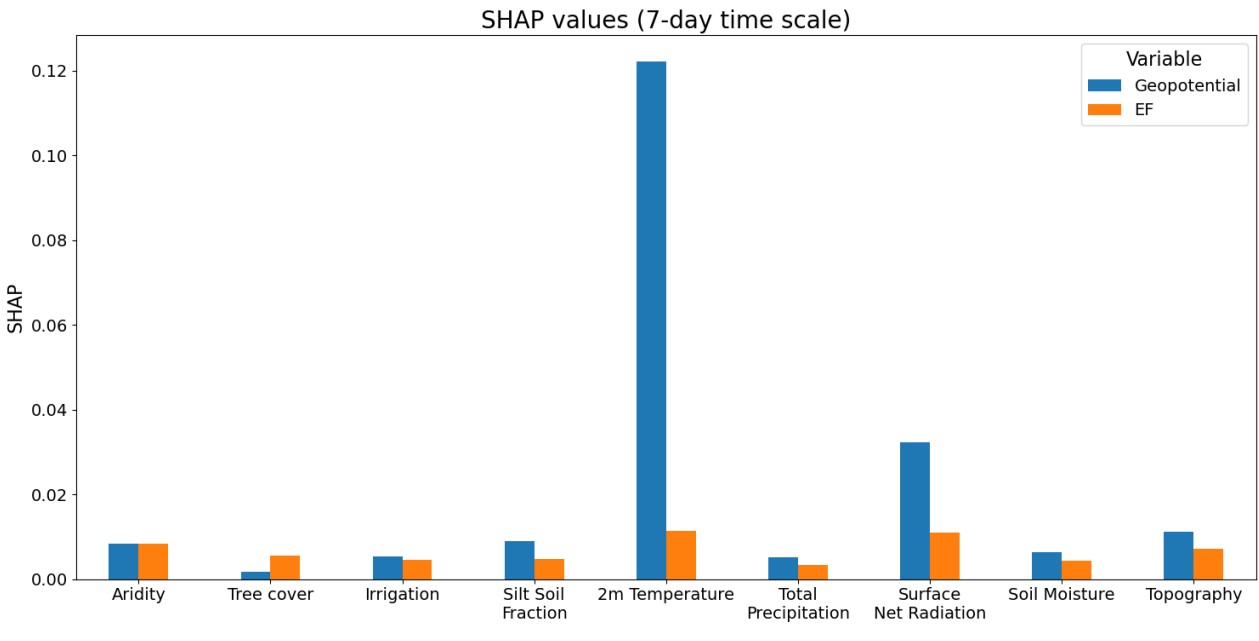

**Figure A3** Relative importance (Shapley Additive Explanations, SHAP values) of multiple factors to explain the spatial patterns of geopotential height and EF as main drivers for 1-day and 7-day hot extremes.

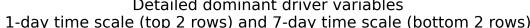

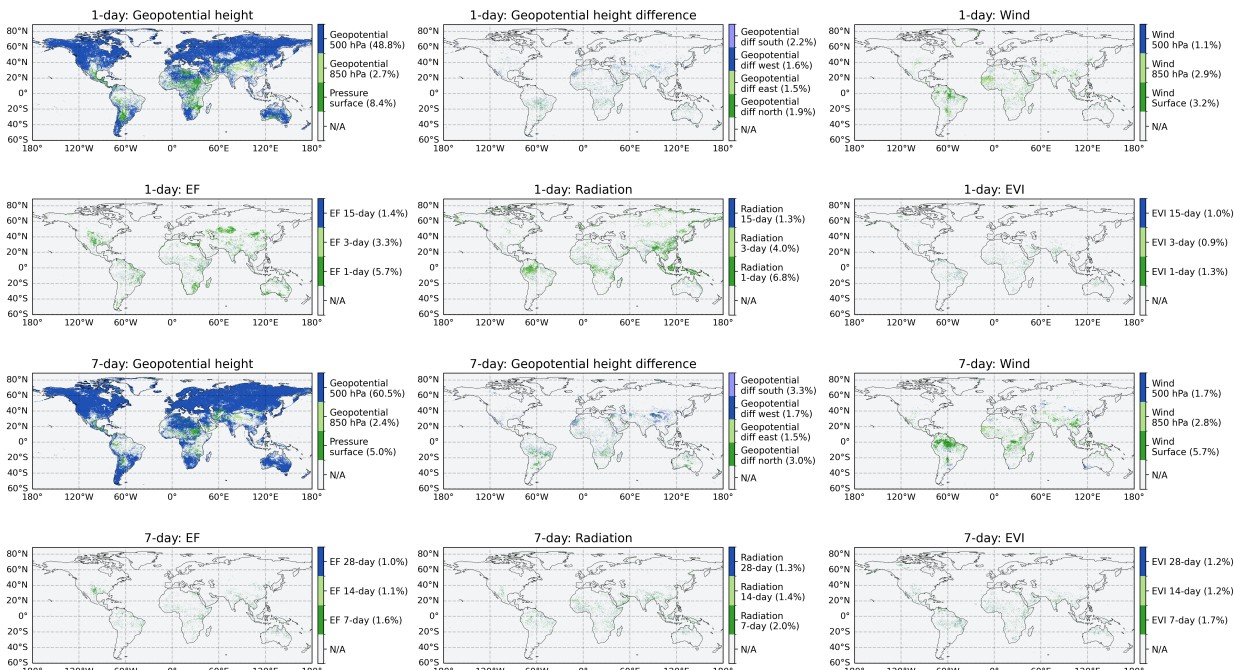

**Figure A4** Detailed dominant driver variables identified for 1-day (top two rows) and 7-day (bottom two rows) time scales are shown for six variable groups: geopotential height, geopotential height difference, wind speed, EF, radiation, and EVI. Each colored grid cell indicates the dominant variable within the respective group. Grey grid cells (N/A) indicate areas where the dominant driver either belongs to a different variable group than the one currently plotted or has missing data. Percentages provided in parentheses on each colorbar indicate the area-weighted fraction

of the total analyzed area where each variable is identified as the dominant driver. These percentages represent the area over which each variable is dominant when considering all variables collectively within each time scale separately. These results are summarized in Table A1.

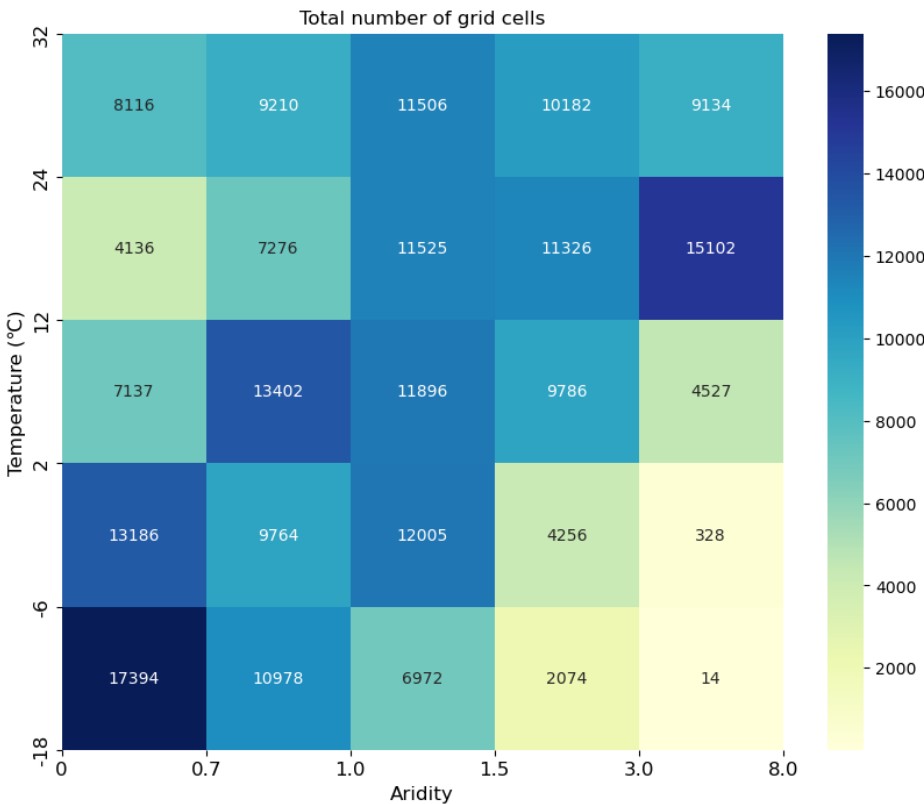

**Figure A5** Total number of grid cells in each temperature-aridity category.

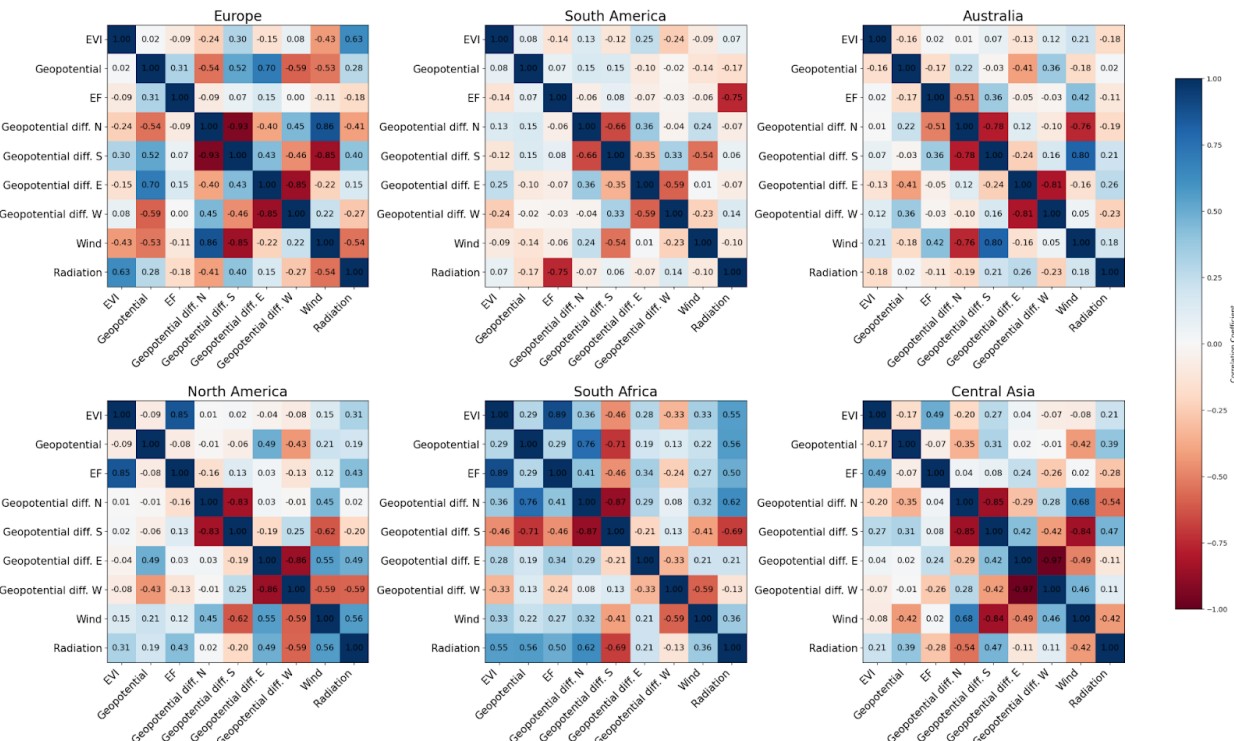

**Figure A6** Spearman cross-correlation matrix for the six land-surface (EVI, EF, radiation) and atmospheric variables (geopotential height, wind, geopotential height difference (north, south, east, west)), averaged across three subregions within each region (See Section 2.5).

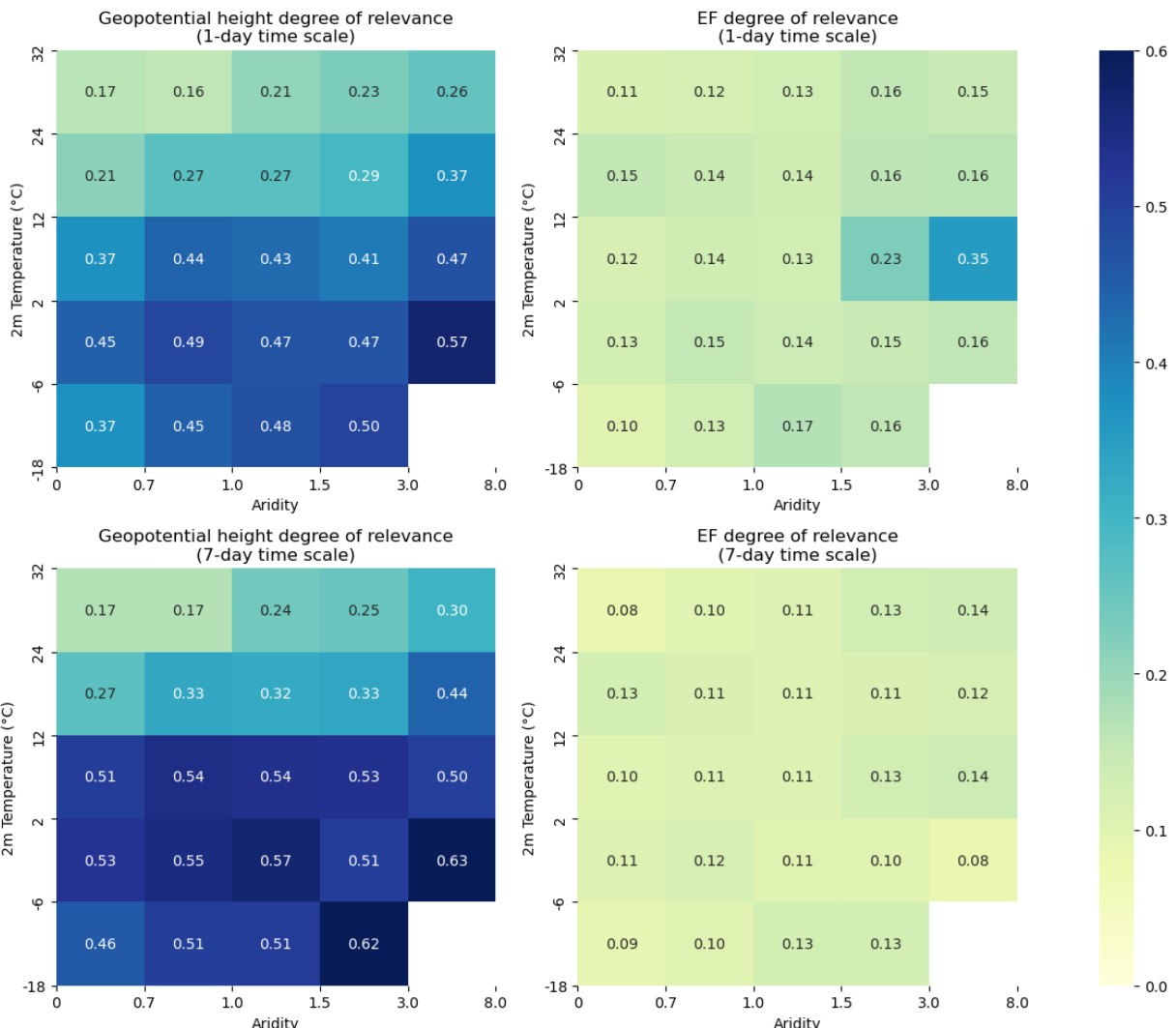

**Figure A7** Geopotential height and EF median degree of relevance summarized across climate classes (given by aridity and temperature ranges) for 1 and 7-day hot extremes. White bins indicate regions that are masked out due to insufficient number of grid cells (less than 20 grid cells)

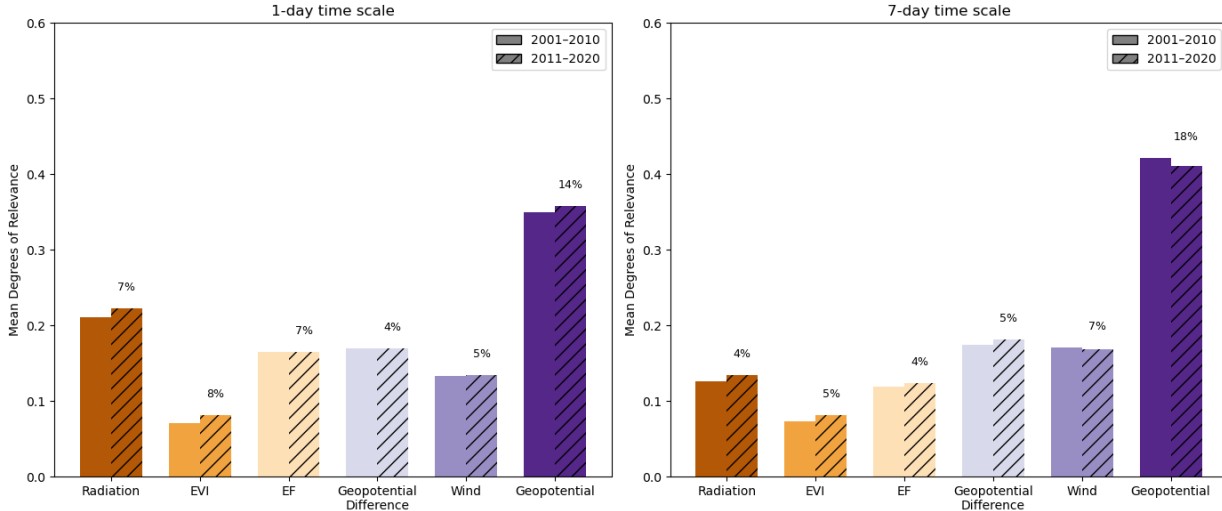

**Figure A8** In the light of the increasing trends in global temperature extremes (Seneviratne et al., 2023), we analyze potential changes in the relevance of the considered drivers of hot extremes over time. For this purpose, we divide the study period into two periods, 2001-2010 and 2011-2020, and employ the same methodology as described in Sections 2.1 to 2.4 to calculate the relevance of all driver variables for each of the time periods. We also determine the significance of the changes in relevance through bootstrapping. For this purpose, we use the original 15 analogues (3 hot extremes x 5 analogues per hot extremes = 15 analogues) per grid cell and per time period (2001-2010 and 2011-2020), and draw a random sample of 15 from them (with replacement). Resampling is done 1000 times. Then, we compare the mean degrees of relevance of the resampled 15 analogues from both periods per grid cell such that we can infer significance from analyzing whether a substantial fraction (e.g. 950 out of the 1000) of the mean degrees of relevance between the two time periods is consistently above or below zero. This figure shows the changes in the mean degree of relevance of the considered hot extreme drivers across the study area between the first and second half of the study period. Bars without hatching denote results for the first half, and bars with hatching show results for the second half. The percentages above each pair of bars indicate the area-weighted fraction of grid cells where driver relevance changed significantly between both decades according to a bootstrap test for significant differences in means with a p-value of 0.05.

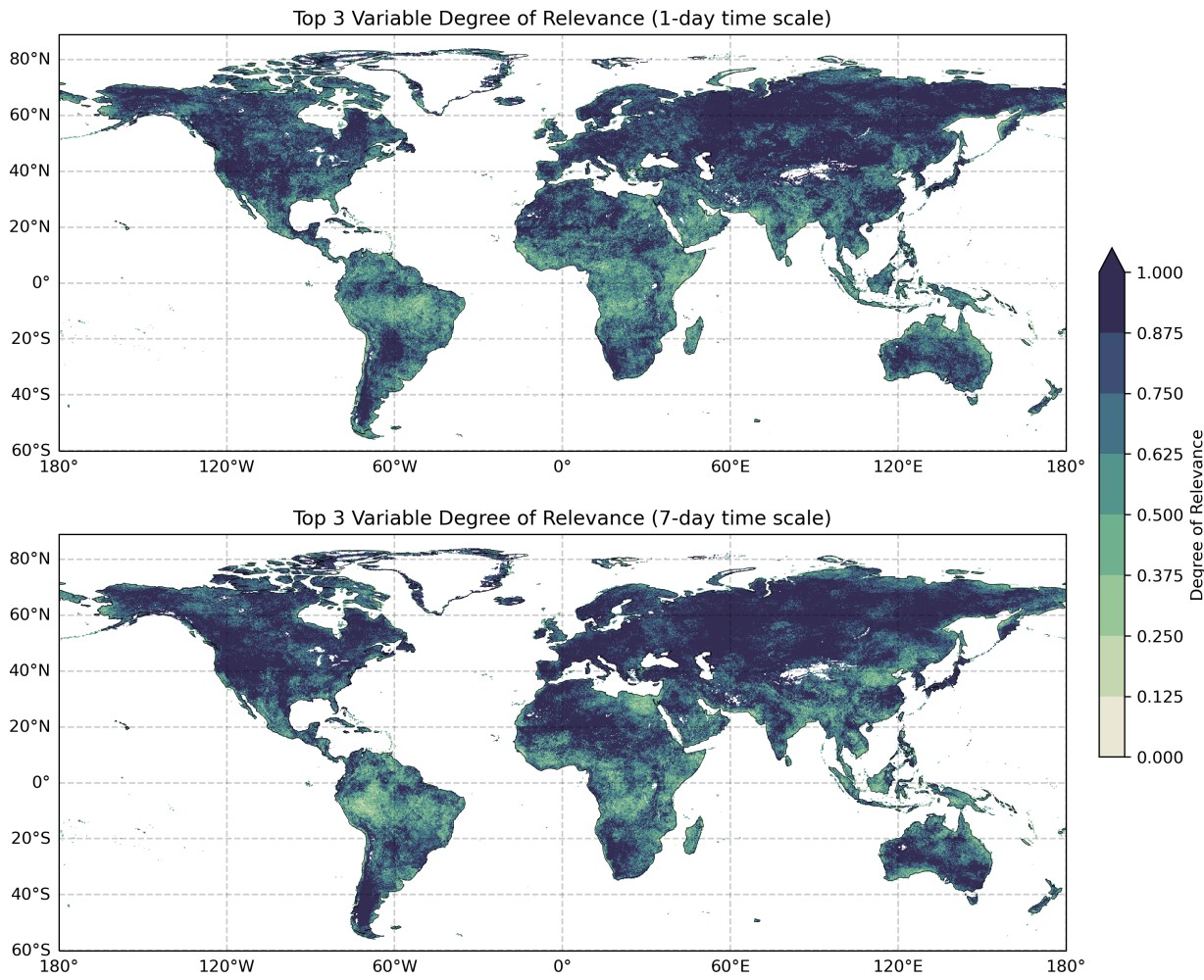

**Figure A9** The sum of the degree of relevance of the three most influential variables for 1-day and 7-day hot extremes. White color indicates ocean/inland water or grid cells with insufficient data.

**Code and data availability**

The variables from ERA5 are available at https://cds-beta.climate.copernicus.eu/datasets/reanalysis-era5-complete?tab=overview (Hersbach et al., 2020). The EVI data from MODIS is available through NASA's data catalogue at https://lpdaac.usgs.gov/products/mod13c1v006/ (Didan, 2015). FLUXCOM-X-BASE evapotranspiration data is available at 445 https://meta.icos-cp.eu/collections/_l85vWiIV81AifoxCkty50YI (Nelson et al., 2024).

**Author contributions**

YU, MRV, RO jointly designed the study. YU performed the computations, plots and data analysis. MRV contributed with coding support and data acquisition. YU, MRV, KDP, RO contributed to the discussion and interpretation of the results, writing of the paper.

**Competing interests**

The authors declare that they have no conflict of interest.

**Acknowledgments**

The authors express their gratitude to Ulrich Weber for his assistance with data retrieval and processing, and to the Hydrology-Biosphere-Climate Interactions group at the Max Planck Institute for Biogeochemistry for their valuable contributions. Melissa 455 Ruiz-Vásquez and Kelley De Polt acknowledge support from the International Max Planck Research School for Global Biogeochemical Cycles (IMPRS). René Orth was supported by the German Research Foundation (Emmy Noether grant no. 391059971). Furthermore, we thank Paul Dirmeyer and an anonymous reviewer for constructive comments.

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
