# Peer review of "Global relevance of atmospheric and land surface drivers for hot temperature extremes"

_EGUsphere, 2024_

## Author Comment (AC1)

Response to reviewers

We are pleased to see that the reviewers value the content of our study. We appreciate their feedback and suggestions. Below, we provide a detailed, point-by-point response to the comments from the reviewers.

Our responses to reviewer comments are organized by category. Each response is labeled with a code in the specified range. The response categories are given below.

| Reviewer Comments | Author Responses |
|---|---|
| CC1 | A1-A2 |
| RC1 | B1-B26 |
| RC2 | C1-C30 |
| CC2 | D1-D5 |

**RC1:**

Review on "Global relevance of atmospheric and land surface drivers for hot temperature extremes" by Yigit Uckan and colleagues.

The manuscript investigates different atmospheric and land surface drivers of hot extremes on two time-scales in the period 2001-2020. The authors find geopotential height to be by far the most important driver of 1 day events, while for longer 7 day events land surface drivers become more important.

The manuscript is well written and well organized and supported by meaningful figures. The topic covered is a timely one and nicely supplements the existing literature, for example, a recent study by Röthlisberger et al. (2023; 10.1038/s41561-023-01126-1) which the authors should consider discussing as it investigates a very similar question but using a quite different approach for atmospheric drivers.

B1: We thank the reviewer for pointing us to this reference and have included it in section 3.1 where we interpret figure 2 (See response B16).

While I do not have any major comments, I see several open questions that should be addressed before publication:

- independence of the variables used as drivers: This is briefly discussed buy should be quantified in some way, in particular to show that atmospheric and land surface drivers are indeed independent as assumed by the authors (line 186).

B2: That is a valid point, thank you for this comment. We believe a cross-correlation matrix can address this. This can be computed by taking 5x5 grid cells to have enough data from the example regions (e.g. central Europe, Amazon, Australia etc.). Since we have 3 hot extreme events per grid cell, this would result with 5x5x3 = 75 data points to compute the correlation between considered driver variables.

Related results will be added to the manuscript as a supplementary figure.

- effect of analogue quality: What role does the 'closeness' of the found analogues to the observed event have on the results and could this influence, e.g., the differences in 1 day and 7 day events (as it might be harder to find good analogues for 7 day events)?

B3: The analogue quality will be assessed by plotting maps of the difference between mean analogue values from the actual values. The variables we will investigate are geopotential, EVI and radiation (as most important predictor variables).

Related maps will be added to the manuscript as a supplementary figure.

- the metric 'degree of relevance' and its interpretation is not quite clear to me as mentioned in the specific comments below

B4: The degree of relevance metric shows to what extent a variable can explain hot extremes in a certain location. It serves as a basis of comparison to determine how much of the observed extreme temperature anomaly can be explained by the analogue temperature conditions of our variables. A detailed explanation is given in our responses C13 and B20.

- the discussion of changes due to climate change (3.3) is very short could benefit from some more analysis and contexualization. In particular since the two time periods investigated are quite short, I'm wondering if any of the effects are statistically significant.

B5: Thank you for this suggestion. We agree that further analysis and contextualization would strengthen this section. Currently, Figure 5 does not indicate whether the observed changes are statistically significant. To address this, we will conduct a bootstrapping analysis for each variable to assess statistical significance.

In this analysis, we will use the original 15 analogues (3 hot extremes x 5 analogues per hot extremes = 15 analogues) per grid cell and per time period (2001-2010 and 2011-2020), and draw a random sample of 15 from them (with replacement). Resampling will be done 1000 times. Then, we can compare the mean degrees of relevance of the resampled 15 analogues from both periods per grid cell such that we can infer significance from analyzing whether a substantial fraction (e.g. 950 out of the 1000) of the mean degrees of relevance between the two time periods is consistently above or below zero.

Identifying statistically significant shifts in relevance will allow a more insightful interpretation of our findings. We will compare these findings with the literature to contextualize changes in relevance. The significance results will be added to Figure 5.

Minor comments
Given that the manuscript is well organized and written this is not a large issue but the authors could consider focusing data section (2.1) better. Currently it reads like a mixture of data and method section, with sentences like: "In addition, we compute the geopotential height

differences at 500 hPa pressure level for each grid cell with respect to the values in adjacent grid cells in the northern, eastern, southern and western directions."

B6: Thank you for your feedback, indeed some sentences belong to the methodology or introduction section. We will revise section 2.1 by moving some sentences to the introduction or methodology parts. Moreover, we will introduce additional subsections in section 2. Specifically, the sections will look like this:

1. Introduction (problem definition, motivation, variables considered in this study, novelty)
2. Data and methods
   2.1 Data (datasets, sources, and preprocessing methods.)
   2.2 Definition of hot extreme events
   2.3 Description of the analogue approach
   2.4 Explanation of the 'relevance index'
   2.5 Description of the attribution analysis
   2.6 Description of the trend analysis

line 79: Could the authors elaborate why the use only 2001-2020?

B7: Evapotranspiration data from the X-base dataset that is used in the calculation of EF is only available between 2001 and 2020. The sentence in section 2.1 has been updated to:

"The spatial and temporal resolutions considered are 0.25 degrees and daily intervals, respectively, for the study period from 2001 to 2020. This period was selected because the evapotranspiration data from the X-base dataset used to calculate EF are only available during these years."

table 1: some of these drivers are probably quite correlated (e.g. GPH and surface radiation), could the authors comment on how this might influence the analysis and interpretation of the results?

B8: We will address this issue by computing the cross-correlations of the variables as we have mentioned in response B2. The correlations will be presented in section 2.1 Data, and we will adapt our discussion in the light of the cross-correlation results.

90: "For the 7-day time scale we apply a moving average" Could the authors state the window size explicitly here? (I'm guessing its 7 days?)

B9: Yes, the window size for the moving average is indeed 7 days. We will revise the sentence in section 2.2 to explicitly state this for clarity.

"For the 7-day time scale, we apply a 7-day moving average to smooth out daily variability."

91: "For each type we select the three hottest events". I'm assuming the authors refer to the 1 day and 7 day events here? However, this is not really clear form the context of the last few sentences.

B10: Yes indeed we refer to the 1 day and 7 day events there. Here we explicitly mention this in section 2.2:

"For the 1-day events, we select three individual days with the next highest temperatures, ensuring that each selected day is at least 15 days apart from the others to maintain independence. For the 7-day time scale, we apply a 7-day moving average to remove variability from shorter time scales and then select the three 7-day periods with the next highest average temperatures, also ensuring they are at least 15 days apart from each other for independence."

To make sure I understand correctly: for the 1 day events these would be 3 individual days and for the 7 day event three 7 day periods?

B11: Yes exactly. We've changed the relevant section in the light of your feedback as mentioned in our response B10.

In general I think this section might benefit from a concrete example. I'm not a 100% sure I understood the approach.
 For example: for the 1 day events, this would give 3 individual days which are all separated by at least 15 days? Meaning if there's a 7 day heatwave only the hottest day of it would be selected for the 1 day event?
Its also never mentioned if this is done on an annual basis (as seems to be indicated in figure 1) or for the entire dataset at once.

B12: Thank you for your comments. The event selection is performed over the entire study period (2001 to 2020), rather than on an annual basis. To ensure clarity, we have updated Figure 1 to better illustrate the process as can be seen in our response B13. We also have added an example to facilitate the understanding of the methodology in section 2.2 in addition to our response B10:

"For example, consider event selection for 1-day hot extremes within the study period (2001-2020). Let's say in a specific grid cell, the hottest day recorded during this period is July 15, 2012. After selecting this day, we mask July 15 and the 30 days surrounding it (July 1 to July 30) to prevent selecting any overlapping or consecutive days. We then identify the second hottest day from the remaining time series after masking, which could be August 5, 2010, and apply the same 15-day masking around this date (July 21 to August 20). This process is repeated to find the third hottest day, ensuring that all three selected days are at least 15 days apart, maintaining their independence.

For 7-day events, the procedure is similar. Suppose the highest 7-day average temperature in the grid cell occurs from July 10 to July 16, 2015. We mask this period and the surrounding 30 days (June 26 to August 1) before selecting the next highest 7-day period, such as August 20 to August 26, 2013. This ensures that each selected 7-day event is independent and focuses on the warm season."

Figure 1:

- could mention that "Hottest Period x" does also refer to a single day in the case of 1 day events?

- increase font size

B13: Thank you for the suggestions. We have updated figure 1 by following your suggestions and the explanation of "Hottest Period x" has been included in the figure caption.

[Figure]

Figure 1 Workflow for determining main drivers of hot temperature extremes. Hottest periods refer to 1-day or 7-day hot extreme events.

"2.5 Effect of the increasing trend" For me 'increasing' indicates an acceleration of a trend (which is already a change measure) so it is probably not what the authors want to say here? 'increasing number' or 'positive trend' instead?

B14: Thank you for pointing this out. We agree that 'increasing trend' might imply an acceleration, which was not our intention. We will revise the section title to 'Effect of the positive trend in hot temperature extremes on the relevance of driver variables' for clarity.

Figure 2:
- the authors could consider using 'more different' colors to make the separation of the different drivers easier? In particular, for the separated view in figure A1 it is almost impossible to really separate drivers due to the chosen colormaps.

B15: Thank you for your suggestion. We understand the challenge you've pointed out regarding the visibility of the drivers in Figure A1. However, due to the high number of variables and the high spatial resolution, achieving clear visual differentiation is quite difficult. Even with adjustments to the colormap, it would be still difficult to clearly distinguish the drivers completely. In Figure 3, we address this issue by highlighting the dominant factors in different climate zones, helping to interpret the results presented in Figure 2. For Figure A1, we will provide a table to summarize the results as we have mentioned in response C26. We appreciate your understanding of these constraints.

- it could also be interesting to compare these results to the work from Roethlisberger et al. 2023 (10.1038/s41561-023-01126-1) at least for the atmospheric drivers?

B16: Thank you for this suggestion. We agree that comparing our results to the findings from Röthlisberger et al. (2023) could provide additional context. Röthlisberger et al. use a Lagrangian approach to decompose temperature anomalies into contributions from advection, adiabatic warming, and diabatic heating, highlighting the regional variability of these processes in forming hot extremes. Findings show that diabatic heating is a factor affecting the temperature anomalies especially in regions where the soil moisture is limited. This finding aligns with our results regarding EF dominant regions. We've added the following sentence to section 3.1 where we interpret Figure 2:

"Similarly, the findings of Röthlisberger et al. (2023) show that diabatic heating is a factor affecting the temperature anomalies especially in regions where the soil moisture is limited. This finding aligns with our results regarding EF dominant regions "

- Röthlisberger, M., Papritz, L.: (2023). Quantifying the physical processes leading to atmospheric hot extremes at a global scale. Nat. Geosci. 16, 210–216. https://doi.org/10.1038/s41561-023-01126-1

153: could this be partly due to the fact that it is (presumably) harder to find good analogues for 7 day events for GPH compared to 1 day events and hence the temperature anomaly is less pronounced for the 7 day case?

B17: We thank the reviewer for raising this point. We will analyze this (see response B3 on the planned approach) and consider these findings also at this place in the results section.

186: "Furthermore, our main goal is to disentangle land surface and atmospheric drivers of hot extremes which are not expected to be strongly related to each other."  As commented earlier: could the authors quantify the cross-dependence of the drivers is some way?

B18: We will address this issue by computing the cross-correlations of the variables as also mentioned in response B2. The correlation methodology will be in section 2.5 and the results will be presented in section 3.1.

192: another limitation might be the quality of the analogues, which seems to be crucial for the quantification of the contribution?

B19: Depending on the outcome of the analysis described in response B3, we will update the limitations section if necessary.  Furthermore, our rationale for analogue selection and the updated relevant text in section 2.3 is given in C11.

Figure 4: "The degree of relevance is computed as the ratio between the respective analogue temperature anomalies and the observed temperature anomalies during hot extremes." This could be explained in a bit more details in the methods section? For example: this seems to mean that the degrees of relevance from different drivers can sum up to more than 100% percent, right?

B20: Thank you for your comment. We agree and will clarify this in the methods section. The degrees of relevance for different drivers can indeed exceed 100% in sum due to collinearity among the drivers, as also shown in Figure A5.

For the methodology part in section 2.4 we've added the following sentence:

"....The expected degree of relevance ranges between 0 and 1. Values closer to 1 can be interpreted as hot extremes are better explained by the relevant variables."

We've added the following sentence to section 3.2:

"....However, in some cases the cumulative degree of relevance of variables can exceed 1 due to collinearities among the variables. Typically, however, this is not the case as shown in Figures 4 and A5. This indicates that dependencies between driver variables are not critically affecting our analysis…."

203: "While EVI is the most relevant driver of hot extremes in more areas at longer time scales (Fig. 2), we find in the main driving variables of hot extremes summarized across climate classes that it also exhibits a higher relevance in these areas but also in other areas where other variables are even more important"
This sentence is somewhat convoluted.

B21: We have updated the sentence in L203:

"Not only the relevance of EVI extends to more regions at 7-day time scale than at 1-day time scale (Fig. 2), we also find that, when summarizing the main drivers of hot extremes across climate classes, EVI's degree of relevance increases in regions where other variables play a more important role."

200-210: This section reads a bit strange in general and seems to make the same point over and over?
"Notably, the relevance of EVI increases with the time scale, in contrast to that of geopotential height, probably due to the longer memory of land surface variables compared to the atmospheric variables"
"This finding highlight that the land surface generally affects hot extremes at longer time scales, as opposed to the more immediate influence of atmospheric drivers."
"This is related to the fact that land surface effects such as evaporative cooling or shading are comparatively smaller but more persistent."
"they are more influential at longer time scales and for hot extremes that build up during a time period without major changes in weather and air masses at a given location"

B22: We agree that the same message has been repeated a couple of times in different parts. We've tried to condensate this into a concise paragraph.

"Notably, the relevance of EVI increases from daily to weekly timescales, likely due to the land surface's 'memory' effects, which allow variables like evaporative cooling and vegetation shading to persist over time. In contrast, atmospheric drivers, such as geopotential height, have a more immediate but shorter-lived influence on hot extremes. This suggests that land surface processes play an important role in driving hot extremes that build up over prolonged periods without major changes in weather or air masses."

215: "Moreover we calculate the sum of the degree of relevance of the three most influential variables at each grid cell (Fig. A5). This shows which part of the observed hot temperature anomalies can be explained with our approach" I think I might misunderstand something here (see also my earlier comment on this). The temperature anomalies from the analogues of different drivers could sum up to more than the observed anomaly, right? So I'm not sure about the interpretation of this.

B23: The reviewer is correct; the anomalies could sum up to more than the observed temperature anomaly (see also response B20). Typically, however, this is not the case as shown in Figures 4 and A5. This indicates that dependencies between driver variables are not critically affecting our analysis. Furthermore, the idea of Figure A5 is to show spatial variations in the explained fraction of the temperature anomalies. We will clarify these points in the revised manuscript in section 3.2 (response B20).

Figure 5: set the maximum of the y-axis to ~.4 to avoid large empty spaces?

B24: Thanks for the suggestion. We have implemented it:

[Figure]

I'd like to see some kind of significance measure for these changes. It seems like apart from EF, none of them are significant even though the authors seem to indicate the opposite in line 233: "At the same time, the relevance of geopotential height, radiation and wind slightly decrease."

B25: We will update Figure 5 with the significance measures for these changes as we mentioned in our response B5.

247: "This finding underscores the significant role of atmospheric blocking mechanisms in the formation of hot extremes" I would assume that most positive GPH anomalies are not blocks even at mid latitudes? In particular on a time-scale of 1 day?

B26: That is a good point, thank you. We can rephrase this sentence:
"This finding underscores the important role of atmospheric circulation anomalies, such as atmospheric blocking, in the formation of hot extremes (Pfahl & Wernli, 2012)."

- Pfahl, S., and H. Wernli (2012), Quantifying the relevance of atmospheric blocking for co-located temperature extremes in the Northern Hemisphere on (sub-)daily time scales, Geophys. Res. Lett., 39, L12807, doi:10.1029/2012GL052261.

---

## Author Comment (AC2)

Response to reviewers

We are pleased to see that the reviewers value the content of our study. We appreciate their feedback and suggestions. Below, we provide a detailed, point-by-point response to the comments from the reviewers.

Our responses to reviewer comments are organized by category. Each response is labeled with a code in the specified range. The response categories are given below.

| Reviewer Comments | Author Responses |
|---|---|
| CC1 | A1-A2 |
| RC1 | B1-B26 |
| RC2 | C1-C30 |
| CC2 | D1-D5 |

**RC2:**

The paper shows an interesting study that teases out how different potential drivers for extreme heat emerge at time scales. The shorter (1-day) time scales are effectively a proxy of the "initial condition" forecast problem, from the atmospheric perspective, in weather prediction, and the strong role of circulation features bears that out. At longer time scales, the "boundary conditions" (i.e., land surface) emerges as an important factor. It brings to mind the point being made in the "infamous" figure used widely in the subseasonal-to-seasonal community (https://www.weather.gov/sti/stimodeling_s2sreport).

The main weakness of the manuscript is a lack of sufficient detail in the description of the methods - I believe this can be easily addressed. The main weakness of the study, as it weakens the conclusions, is the lack of significance testing of the trend analysis. I realize it is not applicable to all the methods shown, but certainly the part of the research comparing changes from the first to second decade of this century could be tested (see specific comments). Otherwise, I think the study has strong merit, and the manuscript can be published after some revisions described below.

C1: We thank Paul Dirmeyer for highlighting the merit of our study, and for the constructive comments below.

General comments:
1. An idea that emerges from this work is validation of the long-held notion that it is circulation features such as stationary ridges that initiate heatwaves (this is clearly stated in a couple of places), but that the land-atmosphere feedbacks (via surface drying and warming – much work by D. Miralles and colleagues on this) can both amplify and prolong heatwaves. The second aspect, prolonging heatwaves, is particularly well demonstrated by this study, and should be emphasized more in the abstract and conclusions, in my opinion. It comes from the novelty of the way a range of timescales has been investigated.

C2: We will include the paragraph below in the conclusion section to highlight the novelty of timescales:

"We reveal that land-atmosphere feedbacks substantially amplify and prolong these events as also shown by Miralles et al (2014). By examining both 1-day and 7-day timescales, we capture different phases of heatwaves—1-day events reflecting the peak of extreme heat, while 7-day events represent prolonged conditions. This approach allows us to infer that atmospheric drivers are likely more relevant in the intensity, whereas land surface drivers, such as surface drying and reduced evaporative cooling, become increasingly important as hot extremes persist."

- Miralles, D. G., Teuling, A. J., van Heerwaarden, C. C., & Vilà-Guerau de Arellano, J. (2014). Mega-heatwave temperatures due to combined soil desiccation and atmospheric heat accumulation. *Nature Geoscience*, *7*(5), 345–349. https://doi.org/10.1038/ngeo2141

We have also updated the sentence in the abstract about the increasing relevance of land surface drivers from daily to weekly time scale as follows:
"The relevance of land surface drivers increases from daily to weekly time scales, supporting the notion that heatwaves are prolonged by land-atmosphere interactions after they are introduced by the atmospheric circulation."

2. Another conclusion that I reached from reading this paper, based on the clear role of EVI (and EF, which is related to canopy conductance that itself links to vegetation carbon uptake and plant processes that regulate that), but that is not made by the authors, is that the results advocate for the inclusion of vegetation phenology in forecast models of weather and subseasonal climate. It is a bit "connecting the dots", but these relationships are arising from processes that are not a part of any operational forecast model (i.e., not parameterized in their land surface schemes), and are even absent from many CMIP models. In the final paragraph of the conclusions, you should point to operational prediction models specifically.

C3: Thank you for your feedback. We will add the modified paragraph to the conclusion.

"….This finding suggests that inclusion of vegetation phenology in operational weather and subseasonal climate forecast models could be crucial, as variables like EVI and EF are linked to vegetation processes, such as canopy conductance and stomatal resistance, which play a significant role in driving hot extremes. Many current forecasting models do not sufficiently exploit the available vegetation data such that they e.g. use only mean seasonal cycles instead of near-real time dynamics. Including these processes would improve the representation of land-atmosphere interactions, which is vital for enhancing the accuracy of hot extreme predictions."

3. Regarding land surface drivers for hot extremes, the literature review is quite short and Euro-centric for a paper with "global" in the title. There are other highly relevant citations in the recent literature that should be noted; a few I am quite familiar with:

https://doi.org/10.1029/2020AV000283,     https://doi.org/10.1175/JCLI-D-20-0440.1,
https://doi.org/10.1175/JCLI-D-22-0447.1, https://doi.org/10.1029/2023WR036490.

C4: We thank the reviewer for pointing us to these references. Here we give a section of the introduction where we emphasize the land surface drivers for hot extremes with more details.

"….On the other hand, land surface feedback mechanisms, including evaporative cooling deficits and vegetation water stress due to low soil moisture can exacerbate the hot extremes and lead to multi-hazard events (Wulff & Domeisen, 2019,  Teuling et al., 2010,  Miralles et al., 2014, Hauser et al., 2016). Similarly, a study by Benson & Dirmeyer (2021) identified a critical "soil moisture breakpoint," below which the probability of heatwaves increases due to a shift in surface energy fluxes from latent to sensible heat. This sensitivity becomes even more pronounced as soil moisture approaches the "permanent wilting point," where vegetation can no longer draw water from the soil, leading to a substantial increase in local surface temperatures. As a result, the sensitivity to soil moisture deficits significantly contributes to the severity of heat events (Dirmeyer et al., 2021). This effect underscores the spatial variability of soil moisture–temperature feedback mechanisms across different climatic zones. Specifically, transitional regions where latent heat flux strongly depends on soil moisture, exhibit more pronounced land-atmosphere coupling (Wehrli et al., 2019; Koster, 2004)...."

Specific comments:
1. L65: It took a while for me to realize that by "height differences" you mean horizontal gradients, relevant to the geostrophic wind relationship. In atmospheric thermodynamics, the term "height differences" is typically applied with respect to the hypsometric relationship, i.e., the vertical distance or "thickness" between two pressure levels, which relates to the mean virtual temperature of the layer between. To avoid confusion, you should replace "height differences" with "horizontal height gradients", or just be explicit that this is a proxy for the geostrophic wind (you could label this as "Advection" here and in Figures 2, 5, A1, A6, A7).

C5: We agree that 'height differences' may lead to confusion. We have replaced 'height differences' with 'horizontal height gradients' in L65.

"In addition, we compute the horizontal geopotential height differences at 500 hPa pressure level for each grid cell with respect to the values in adjacent grid cells in the northern, eastern, southern and western directions."

2. §2.1: It is stated that daily data (shortest time scale) are used. Are these based at each point on the local time, or all on 0000UTC as the day boundary? If the latter, then for about half of the world, what you call "one day" actually spans two days with respect to important diurnal phase of drivers like net radiation and evaporative fraction. Please clarify and/or justify the choice.

C6: Thank you for the valuable feedback. We confirm that the ERA5 daily means used in our analysis are computed based on UTC+00:00. We acknowledge that this approach may lead to phase mismatches in diurnal cycles for some variables, particularly in regions where local time differs significantly from UTC (Zou and Qin, 2010). However, we believe this does not substantially affect our results, as the drivers in our analysis do not change significantly from one day to the next, and the diurnal cycle is inherently accounted for when using daily means.

We have clarified this in L51:
"The daily means used in our analysis are computed based on UTC+00:00. While this choice may lead to phase mismatches in diurnal cycles for some variables, particularly in regions where local time differs significantly from UTC, it provides consistency across datasets, which is essential for our analysis."

- Zou, X., & Qin, Z.-K. (2010). Time zone dependence of diurnal cycle errors in surface temperature analyses. Monthly Weather Review, 138(6), 2469–2475. https://doi.org/10.1175/2010mwr3248.1

3. L70-71: This is the first time either "X-BASE" or "ERA5" are mentioned (before Table 1 is cited). They should be defined or described here, or else moved after Table 1.

C7: Here's the revised sentence in L70-71:
"It is important to note that we compute EF using variables from two different datasets: X-BASE and ERA5. This approach is justified, as X-BASE is formulated using ERA5 data."

4. Table 1, EF: How is EF calculated from X-BASE? To my knowledge, the publicly released data does not contain this variable, nor the necessary data to calculate it (i.e., there is no sensible heat flux field). That renders this part of the study unreproducible by others.

C8: Thank you for your question. The relevant calculation is explained in line 69 (section 2.1):

"EF is computed by normalizing evapotranspiration (ET), which we retrieve from X-BASE, by surface net radiation, which we retrieve from ERA5."

5. L92: How are the warm seasons defined? There are a number of approaches, as there are strong latitudinal (and more complex) determinants. Are the same number of months used everywhere, or are only very cold months avoided (a temperature threshold)?

C9: The warm season for each grid cell is defined by identifying the hottest day based on absolute temperature and then applying a 60-day window centered around that day. This approach is repeated for all years within the study period, ensuring consistency in the number of days considered as the warm season for each grid cell. Consequently, each grid cell has the same number of days defined as the warm season across all years, based on local temperature patterns.

6. Figure 1: This is an important figure, but it not clear and the descriptions in the text do not fully clarify the workflow, especially for part (c). In the caption, it should explicitly say "see text for details".

C10: Thank you for your comment on Figure 1. We appreciate your feedback, and since similar points were raised by the other reviewer (see response B10), we have made adjustments to improve the clarity of the figure and the workflow description, especially for part (c). The revised figure and accompanying text now provide a clearer explanation of the workflow. We have also included 'see text for details' in the caption as suggested.

7. L100: Please give more description of the definition of "similar" (i.e., please do not rely solely on a reference to Yiou et al. 2007). Is it based on RMSE? Is some normalization applied? Perhaps it is best to include equations.

C11: We identify analogue periods by selecting the five periods with driver values most comparable to those observed on the hot days based on one dimensional euclidean distance. We have added additional explanation to the manuscript to describe our selection process in detail. This clarification has been added to section 2.3.

"....This means that for each driver and at each considered atmospheric level (i.e., geopotential height and wind) and temporal scale (i.e., EVI, EF, and surface net radiation) we select the five periods with the raw driver values most similar to those observed during the identified hot extremes based on one dimensional euclidean distance. This approach shares a conceptual basis with the analogue methods in the literature, such as those used by Jézéquel et al. (2018) and Yiou et al. (2007). These studies show that selecting more than five geopotential height analogues has little significant effect on the results. Also to maintain consistency across all grid cells, we use the same number of analogues in our analysis."

- Jézéquel, A., Yiou, P. & Radanovics, S., (2018): Role of circulation in European heatwaves using flow analogues. Clim Dyn 50, 1145–1159 . https://doi.org/10.1007/s00382-017-3667-0
- Yiou, P., Vautard, R., Naveau, P., & Cassou, C., (2007): Inconsistency between atmospheric dynamics and temperatures during the exceptional 2006/2007 fall/winter and recent warming in Europe. Geophys Res Lett, 34(21), https://doi.org/10.1029/2007gl031981

8. L103: This is not entirely clear – do you mean that the center of the window is on the calendar date (month and day) applied across all years?

C12: The ±60-day terminology we used in the text actually refers to a 120-day window around the specific calendar date (month and day) of the hot extreme event, applied across all years to select the analogue periods. This approach ensures that each analogue period is centered on the same seasonal timing as the event while maintaining at least 15 days of separation to ensure independence. We acknowledge that the phrasing is not clear here. We have revised the sentence in section 2.3.

"....For this purpose, a 120-day window surrounding the specific calendar date (i.e., month and day) of the relevant hot extreme event is considered across all years to select the analogue periods. These selected periods are also at least 15 days apart from each other to ensure independence."

9. §2.4: As noted above, this description is very fuzzy. I do not follow the process. Again, perhaps equations or pseudocode is needed. I would not be able to reproduce this methodology based on the description.

C13: We've added an equation in section 2.4 to better explain the methodology:

"For each variable $D$ (e.g., geopotential height, EVI, EF), we identify 15 analogue periods based on similarity to the three hottest observed extremes (5 analogues each) in each grid cell. The degree of relevance for each variable "D" in each grid cell "g" is computed as follows:

$$Degree\ of\ relevance(g, D) = \frac{1}{15} \sum_{i=1}^{15} \left( \frac{T'_{analog}(g,i)}{\bar{T}'_{event}(g)} \right)$$

T'_analog(g, i) denotes the temperature anomaly during the i-th analogue period in grid cell g, based on the conditions of driver D.
T'_event(g): This is the mean temperature anomaly calculated from the three observed hottest extreme events in the grid cell g. It serves as a basis of comparison to determine how much of the observed extreme temperature anomaly can be explained by the analogue temperature conditions of variable D."

10. L114-115: This is, of course, a linear assumption, that the drivers can be considered separately. This point is acknowledged later as a possible drawback, but it would be good to state that here – this is where some readers will begin to have this question in their minds.
C14: We've added the following lines:
"It is important to note that this approach assumes a linear and separate contribution of each driver, which may be a limitation when interactions between drivers are relevant."

11. L120: Should "both" be replaced with "each of the"?
C15: We have changed the wording as you suggested:
" For this purpose, we divide the study period into two periods, 2001-2010 and 2011-2020, and employ the same methodology as described in Sections 2.1 to 2.4 to calculate the relevance of all driver variables for each of the time periods."

12. Figures 2, 4, A1, A5, A7: It is difficult to tell the grey from some of the pale blue shades – they have very similar luminance. Additionally, the monochrome palettes in Figures 4 and A5 make them somewhat hard to read. It appears that you are trying to be

considerate of colorblind readers – using a cubehelix palette in these two figures would improve clarity for all.

C16: We will change the background color to a darker gray to improve contrast with the pale blue shades. We will also use a cubehelix palette with varying luminance in Figures 4 and A5 to enhance readability and ensure accessibility for colorblind readers.

13. L142-143: Please move this final sentence of the paragraph up to become the 2nd sentence (right after Fig. 2 is mentioned).

C17: The sentence is now in line 128:

"The global distribution of the dominant variables for both 1-day and 7-day time scale extreme temperatures are illustrated within Fig. 2. A more detailed depiction of drivers' relevances across height levels and time scales is presented in Fig. A1."

14. L154-163: This is methodology: it should be explained in §2, not with the results. Additionally, how the Random Forest method is applied must be explained in sufficient detail such that a reader could hope to reproduce it.

C18: The relevant section will be moved to section 2.1 and updated as follows:

"In order to analyze the spatial distribution of the dominant driver variables identified for 1-day and 7-day hot extremes with respect to different land surface characteristics and climatic regimes, we employ a random forest approach where geopotential height and EF serves as target variables while a range of hydro-climatological, vegetation and landscape variables is used as predictors. The data were processed by separating the target variables from the predictors and splitting them into training and testing sets, with 25% of the data reserved for testing and a random state of 42 to control the shuffling applied to the data before applying the split. We used 100 trees, and a maximum depth of 10 to configure the RandomForestRegressor, as these hyperparameters have proved to work well in other studies (Oshiro et al.; 2012; Probst & Boulesteix 2017). Bootstrapping was enabled, and the feature importance was evaluated using SHAP (Shapley Additive Explanations) values to provide insight into the contribution of each predictor. The mean absolute SHAP values were calculated, and we found that long-term mean temperature and radiation are the most relevant predictor variables for both 1-day and 7-day hot extremes. Additionally, aridity (calculated as the ratio of long-term mean net radiation and unit-adjusted long-term mean precipitation) and topography play a role while the other considered variables are less important. While temperature has the highest relevance and is therefore selected as the primary variable, radiation, which ranks second in relevance, is closely related to temperature as an atmospheric variable. To ensure the inclusion of a land surface-related factor, we choose aridity, which captures the interaction between radiation and precipitation, thus providing a metric for assessing land surface influences on hot extremes. (Fig. A2)."

- Oshiro, T. M., Perez, P. S., & Baranauskas, J. A. (2012). How many trees in a random forest? In Machine Learning and Data Mining in Pattern Recognition: 8th International Conference, MLDM 2012, Berlin, Germany, July 13–20, 2012, Proceedings, Vol. 7376, 154. Springer.

- Probst, P., & Boulesteix, A.-L. (2017). To tune or not to tune the number of trees in a random forest? Journal of Machine Learning Research, 18, 1−18.

15. L168: Replace "mostly just" with "barely".

C19: "Water availability is barely sufficient for vegetation in these regions, which means that (i) it can provide significant evaporative cooling; however, (ii) during warm and dry conditions, the limited water availability becomes insufficient, leading to reduced evaporative cooling and consequently enhanced temperatures.

16. Figures 3, A3, A4, A8: Aside from aridity=1.0, which has a special meaning in the Budyko framework, the other boundaries for the bins do not necessarily need to be chosen because they are round numbers or evenly spaced. If instead you had chosen boundaries on each axis that contained approximately equal numbers of grid cells, you may arrive at a more robust and clear result with fewer dependencies on varying sample sizes. But I would suggest keeping a boundary for aridity at 1.0.

C20: Thank you for the suggestion. We have tried to use different aridity and temperature classes to obtain approximately equal numbers of grid cells in each bin as can be seen from the heatmaps below. We will update the aridity binning as shown in the first-row plots in order to have more similar number of grid cells in each box, and apply this binning for all heatmap visuals we use in the manuscript. The second row shows the previous version that we had in our manuscript.

[Figure]

[Figure]

[Figure]

**Figure A3** Total number of grid cells in each temperature-aridity category

17. L181-192: I appreciate this paragraph. If you are interested in pursuing this further, you might consider using an approach based in information theory, which has the advantage of also being nonparametric. There are also ways to quantify nonlinearity and parameter interaction (see: https://doi.org/10.1002/2016WR020218, https://doi.org/10.1002/2016WR020216, https://doi.org/10.1029/2020WR028179).

C21: Thank you for this suggestion. We appreciate the reference to information theory and its advantages. However, at this stage, pursuing this further in that direction would extend beyond the current scope of our paper, but we acknowledge that information theory-based methods could be an interesting approach for future studies.

18. L190: I think the independence of different data sources could be looked upon as a strength, not a weakness, of this research. When patterns emerge across datasets with different algorithms, or not all from one model, it gives more credence to the results.

C22: Thank you for this comment. We agree that the use of independent data sources can indeed be viewed as a strength of our study. We will mention this also in the discussion section. Here's a revised version:

"Another limitation is the data quality of each driver variable. A lower signal-to-noise ratio for certain variables compared to others may affect the identification of analogues and related temperature anomalies, and consequently the estimated relevance of the variable. However, the use of independent data sources can also be considered a strength of our study. We observe consistent patterns across different datasets which enhances credibility to our results and align with the existing literature on land surface and atmospheric patterns."

19. L205: Replace "highlight" with "highlights".

C23: "This finding highlights that the land surface generally affects hot extremes at longer time scales, as opposed to the more immediate influence of atmospheric drivers."

20. Figure 5: The result is not compelling unless statistical significance of these differences between decades can be established. Fortunately, that is straightforward. A very robust test is a bootstrap approach where the 20 years are randomly split into 2 sets of 10 and

the "degree of relevance" calculation is repeated many times (say 1000 times; C(20,10) = 184756, so no problem with oversampling). Then find where the particular case of 2001-2010 versus 2011-2020 falls in the larger distribution… that is your p-value. Otherwise, we don't know if the EF changes are meaningful.

C24: Thank you for suggesting a method for the statistical significance of figure 5. We will implement a bootstrapping analysis as suggested. See response B5 for details on the planned approach.

21. L245: Drop the word "wide" – it's not very appropriate.

C25: We have revised the sentence to read: "This study provides a comprehensive analysis of the potential drivers of hot extremes, considering a selection of atmospheric and land surface variables."

22. L246: You say "particularly at the 500 hPa level" but the results for other levels were never quantified, save for squinting at all the similar shades of color in Figure A1. A Table (A1, perhaps?) should be included with the complete quantifications (for all factors in each decade – as Figure A7 is also difficult to read).

C26: Thank you for the suggestion. We will include a table in Appendix A that provides a complete quantification of the variable percentages for all factors at each level and for each decade.

23. L267: Here you are talking about trends, but you do not use the word "trend". It would be clearer if you did.

C27: We have revised the text to include the word 'trend' for clarity.
"Another interesting result of our study is the positive trend in the relevance of the land surface in general, evaporative fraction in particular, driving hot extremes during the study period. This is likely related to higher temperatures and precipitation variability, which enhance the role of evaporation in the surface water and energy balances."

24. Figure A2: Please expand the acronym "SHAP".

C28: We will expand the acronym 'SHAP' in the figure caption to improve clarity.
"Relative importance (Shapley Additive Explanations, SHAP values) of multiple factors to explain the spatial patterns of geopotential height and EF as main drivers for 1-day and 7-day hot extremes."

25. Figure A7: I suggest for the bottom 2 panels, only color the grid cells where a change has occurred. Leave the unchanged cells blank.

C29: Thank you for the suggestion, we will revise the figure accordingly.

26. Figure A8: Presumably there is a bit of movement of some grid cells between bins from one decade to the next. Are you considering that here, or are the 2m temperature and aridity still based on the 20-year climatology? Also, here again, a bootstrap statistical test can tell which bins have significant changes (or perhaps use color to indicate p-value).

C30: We are not considering decadal changes in aridity and mean temperature, but use the 20-year averages to create the bins. We will test to do the binning for each of the two 10-year periods to see if there are noteworthy changes in the number of grid cells per bin.

---

## Author Comment (AC3)

Response to reviewers

We are pleased to see that the reviewers value the content of our study. We appreciate their feedback and suggestions. Below, we provide a detailed, point-by-point response to the comments from the reviewers.

Our responses to reviewer comments are organized by category. Each response is labeled with a code in the specified range. The response categories are given below.

| Reviewer Comments | Author Responses |
| --- | --- |
| CC1 | A1-A2 |
| RC1 | B1-B26 |
| RC2 | C1-C30 |
| CC2 | D1-D5 |

**CC2:**

Dear authors
The theme of this article is extremely interesting and important to me.

D1: We are happy to see that our work is recognized and of interest.

However, I am not acquainted with the data you use, nor with the techniques you use. So, my comments below may be completely irrelevant.
However, maybe I misunderstood, but I anticipated that you i) would show where hot extremes could occur under increasing global warming (in terms of temperatures, K, and region, that is, a Figure 2, but with colors showing extreme temperatures. Second, ii) I anticipated the map you show in Figure 2 of the possible causes for the hot extremes, but I assumed that you would have included the effects of ocean temperature variability. Ocean variability seems to have played a dominant role for global warming until about 1950, that is, the cold phase in ocean variability could compensate for increases in CO2, e.g., Wu et al. (2019).

D2: Thank you for pointing out these interesting aspects.
i) While mapping extreme temperatures under increasing global warming would indeed be interesting, our study is focused on the global distribution of drivers of hot extremes rather than on the spatial distribution of the extreme temperatures themselves. Our analysis examines the drivers independently of the temperature values associated with each hot extreme event. However, we do categorize the relevance of different drivers by region, as shown in Figure 3, which aligns with a similar intent to understand the spatial variation of hot extreme drivers.
ii) Ocean variability is indeed an important aspect of hot extremes. However, as we mentioned in our response A1, we only consider land surface and atmospheric drivers of hot extremes. Incorporating ocean variability indices in future analyses could provide valuable insights into how large-scale oceanic patterns influence continental temperature extremes. We will acknowledge this fact in our introduction.

Also, maybe I am too numerical, but for me an equation like
T = a1 geopotential (unit)  + a2 wind (m.s-1) +..
with variables centered and normalized to unit standard deviation to avoid any effect of the units.  Since I am not sure the variables are "strongly  not related" line 189, maybe a Principal component analysis, PCA, would be appropriate (I don't know).

D3: Thank you for the suggestions. We will calculate a cross-correlation matrix to quantify the correlations of the variables. See response B2 for details on the planned approach.

You use the term "Dominant driver" , "… while net radiation is the dominant driver in a slightly larger area.." but I am not sure how you come to that conclusion, except that it covers a larger portion of a study area.

D4: In our study, we use the term 'dominant driver' to describe drivers that are found to have the strongest influence across the largest area of the study region. This is based on the spatial extent where each driver is most relevant, rather than on a direct quantitative comparison of their intensities or magnitudes. As a result, when we describe net radiation as a 'dominant driver,' we mean that the area it covers as the most influential driver is larger for 7-day hot extremes compared to 1-day hot extremes.

"We find that long-term mean temperature and radiation are the most relevant predictor variables for both 1-day and 7-day hot extremes " , and I am not sure what "Most relevant" means. I would have anticipated some numerical values here.
If my comments do not give any meaning t you, please just skip them.

D5: We used SHAP (Shapley Additive Explanations) values to provide insight into the contribution of each predictor, in other words to determine the relevance of these features. The mean absolute SHAP values were calculated, and we found that long-term mean temperature and radiation are the most relevant predictor variables for both 1-day and 7-day hot extremes. The numerical values of the SHAP values are given in Figure A2. A detailed explanation of how we calculate the relevance (SHAP values) and the updated text is given in our response C18.

Best Knut L. Seip
Wu, T. W., Hu, A. X., Gao, F., Zhang, J., & Meehl, G. A. (2019). New insights into natural variability and anthropogenic forcing of global/regional climate evolution. Npj Climate and Atmospheric Science, 2. https://doi.org/UNSP 18
10.1038/s41612-019-0075-7

---

## Author Response (AR2)

The manuscript is much improved. But I am now finding that I am unsure about some aspects of the procedures (see below). I think this can be clarified rather easily with some improvements to the description of the methodology. Also, I think the material on climate trends can be removed – it is not contributing to the main goals of the paper. Otherwise, specific comments are given below.

A1: We thank Paul Dirmeyer for his additional feedback on our study.

General comments:
1. I think the part on climate trends (Sections 2.7, 3.3 and Figure 5) can be dropped. With only 20 years of data considered, the likelihood of separating signal from "noise" (interannual to decadal variability) is slim. The method chosen, comparing statistics calculated from two adjacent decades, which necessarily have first moments separated by only 10 years, unsurprisingly finds nothing. There are other approaches such as those that focus on trend detection in time series that would have been more efficacious, but probably still would not produce significant results with fewer than 30-40 years of data. Furthermore, the reasons given for what are insignificant trends are themselves only speculative. Unless the authors feel compelled to include climate change because it was the topic of the funding grant, for instance, I think the paper would be better without this distracting section.
A2: We agree that the part on climate trends may be less robust than our other main results. For this reason, we decided to move the trend results into the appendix (former Figure 5 is now Figure A9 and former section 2.7 is now the description of Figure A9) such that in the main text we only shortly summarize the main findings of the trend analysis in lines 297-301:

"Furthermore, we study potential changes in the relevance of the considered drivers of hot extremes between the periods 2001-2010 and 2011-2020 (Fig. A9). We find only small changes. While these could be related to natural decadal variability, we find that radiation and EVI are becoming slightly more relevant at both considered time scales, which may be also related to global greening (Zhu et al., 2016, Chen et al., 2019) and global brightening (Wild, 2009). At the same time, the changes are significant only in relatively small fractions (<10% for most drivers) of the study area."

2. As the description of the methodology in section 2.3 has been clarified, I realize there is an important detail that was not mentioned. Figure 1 suggests that a separate set of analogues is determined independently for each variable (geopotential, EF, etc.). So, over what spatial domains are analogues determined? I understand that goodness-of-fit is determined by RMSE (Euclidian distance), I think this must be between 2-D fields, so is the domain always global, or is it some radius centered on each grid cell in question? It seems that the analogues should be regional in nature. Or am I completely misunderstanding the procedure? This is causing me to doubt my understanding of Figure A1 as well – is each map a composite of the difference at each grid cell for each grid cell's spatial analogue? But then Euclidian distance

should always be positive, and we have negative values too, so no I am unsure of the calculation behind Figure A1.

A3: Thank you for pointing this out. Actually, analogues are determined independently for each variable in each grid cell. So, the analogues are from the same spatial domain (same grid cell).

Euclidian distance should result with positive values; however in our case, we only subtract the analogue values from the observed value, which can result in positive or negative differences.

We have removed the one dimensional Euclidian distance term in line 137, and have added a sentence to clarify that analogues come from the same grid cell where the hot extreme was identified in lines 142-143:

"The analogue analysis is done separately for each grid cell, i.e. analogues always come from the same grid cell where the hot extreme was detected."

Specific comments:

1. Abstract: "Analogues" are the tool of choice for this research, but the term is not mentioned in the abstract. Something about the methodology should be mentioned in the abstract.

A4: The term was actually mentioned in the abstract, but not explained. We have now expanded this part of the abstract a bit in lines 15-16:

"Hot extremes are identified at daily and weekly time scales through the highest absolute temperature, and the relevance of the considered drivers is determined with an analogue-based approach. Thereby, temperature anomalies are analyzed from situations with driver values similar to that of the hot extreme."

2. L128: Missing period.

A5: Done

3. Fig A1: This shows percentage differences, but that doesn't work well when, for some variables, the mean is much larger than the variance (e.g., 500hPa geopotential height). Wouldn't it make more sense to normalize the differences, e.g., locally by the overall (across 20 years) standard deviation of the field at each point?

A6: We agree that normalizing would solve this issue. The differences are now normalized by the local (20-year) standard deviation. We have updated Figure A1 with the figure below.

[Figure]

Figure A1. Spatial patterns of the normalized difference of geopotential height, radiation and EVI between the mean values of the five analogue periods and the values of the observed hot event divided by the local (20-year) standard deviation. Note that this figure shows the average differences across the three hottest periods. Mean and standard deviation values are denoted in the bottom left corner of each map

4. L177: change "cells" to "cell"
A7: Done

5. L192: I am not familiar with SHAP. Is there some threshold for significance that can be determined? What SHAP value would noise produce? Looking at Fig A3, I have no intuition of what values are important.

A8: SHAP values are a way to interpret the relative importance of each predictor for each target variable. In our case, aridity, tree cover, surface net radiation, soil moisture, and topography are among the predictors used, while geopotential height and EF serve as target variables. Since SHAP values only indicate the relative importance of each predictor, there is no significance threshold. Therefore, the importance of a SHAP value is best understood by comparing it to the values of other predictors within the same model.

We additionally plotted the same figure including a noise variable (see Figure R below), which was generated from a random normal distribution. This figure demonstrates that the noise predictor makes almost no contribution to explaining geopotential height and EF compared to the other predictors.

[Figure]

[Figure]

Figure R. Relative importance (Shapley Additive Explanations, SHAP values) of multiple factors to explain the spatial patterns of geopotential height and EF as main drivers for 1-day and 7-day hot extremes. Additionally, a noise predictor is added, which is an artificial data with a normal distribution.

6. L 291: Change "In a next step, we are determining…" to "In the next step, we determine…"
A9: Done

7. L296: Change "are grouping" to "group".
A10: Done

8. Table S1: I believe this should be in reference to Figure A4, not A2. However…
A11: Done

9. Figure A4: The colors are all so similar that it is impossible to tell much about spatial distributions beyond the purple vs brown. Only the percentages next to the colorbars are informative, and that information is in Table A1. With so many gradations (20 categories), it is not possible to produce readable high-resolution maps. If this information is vital to portray in map form, the authors need to consider a different method – it may require 3 to 6 maps for each time scale, with one or two variables (among six sets listed) per map. Otherwise, maybe only the table is necessary.
A12: We appreciate the reviewer's feedback. To improve clarity and readability, we have revised Figure A4 by reducing the complexity of the maps. Specifically, we plotted each variable group (e.g. geopotential height at different pressure levels) resulting with 6 maps for each time scale. This adjustment enhances the distinction between different variable groups and makes the spatial distribution patterns easier to interpret visually.

[Figure]

Figure A4. Detailed dominant driver variables identified for 1-day (top two rows) and 7-day (bottom two rows) time scales are shown for six variable groups: geopotential height, geopotential height difference, wind speed, EF, radiation, and EVI. Each colored grid cell indicates the dominant variable within the respective group. Grey grid cells (N/A) indicate areas where the dominant driver either belongs to a different variable group than the one currently plotted or has missing data. Percentages provided in parentheses on each colorbar indicate the area-weighted fraction of the total analyzed area where each variable is identified as the dominant driver. These percentages represent the area over which each variable is dominant when considering all variables collectively within each time scale separately. These results are summarized in Table A5.

10. Figure A6: I do not understand how Wind and Geopotential difference can be strongly anticorrelated in Australia, but positively correlated elsewhere. Is there perhaps some factor in calculating geopotential difference that changes sign in the Southern Hemisphere (e.g., the Coriolis term) that creates this inconsistency? But then why is it not also present for South Africa and South America? I find this puzzling.

A13: We previously used only northward geopotential height differences in that figure, but we have now included all other geopotential height differences. In this updated figure, we observe that the negative correlation identified earlier might be related to a strong positive correlation in the opposite direction, influenced by local conditions (e.g., aridity and topography) specific to this site and not representative of other Southern Hemisphere locations. Furthermore, since the geopotential height differences are computed at 500 hPa while wind speed is measured at the surface, we suggest that a strong geopotential height gradient at 500 hPa does not necessarily translate to strong surface wind speeds. This discrepancy could be attributed to local land surface characteristics such as topography and vegetation cover.

[Figure]

Figure A7 Spearman cross-correlation matrix for the six land-surface (EVI, EF, radiation) and atmospheric variables (geopotential, wind, geopotential difference (north, south, east, west)), averaged across three subregions within each region (See Section 2.5)